# Autocorrelation Test under Frequent Mean Shifts

**Abstract**

Testing for the presence of autocorrelation is a fundamental problem in time series analysis. Classical methods such as the Box-Pierce test rely on the assumption of stationarity, necessitating the removal of non-stationary components such as trends or shifts in the mean prior to application. However, this is not always practical, particularly when the mean structure is complex, such as being piecewise constant with frequent shifts. In this work, we propose a new inferential framework for autocorrelation in time series data under frequent mean shifts. In particular, we introduce a Shift-Immune Portmanteau (SIP) test that reliably tests for autocorrelation and is robust against mean shifts. We illustrate an application of our method to nanopore sequencing data.

**Keywords:** Box Test, Nanopore Sequencing Data, Non-stationarity, Portmanteau test, Quadratic Estimator, Time Series.

**Mathematics Subject Classification (2020):** 62M10

## 1  Introduction

Testing for serial correlation has been a fundamental and classical problem in time series analysis. The most celebrated Box type portmanteau tests (Box and Pierce, 1970; Ljung and Box, 1978) and their variants (see for example Escanciano and Lobato, 2009, for a review) have become common practice for decades. This paper considers the white noise tests in the presence of frequent change points, defined as times or locations when the mean of the process shifts. Most of the existing literature on change-point analysis have been concerned with the detection of change points, see Niu et al. (2016) and Truong et al. (2020) for some overviews. However, when the temporal dependence is present, it has to be taken into account by any detection procedure (Tang and MacNeill, 1993). While the model-based approaches (Davis et al., 1995; Gombay, 2008) account for it intrinsically, the problem becomes more sophisticated for nonparametric methods. Altissimo and Corradi (2003); Juhl and Xiao (2009); Shao and Zhang (2010); Aue and Horváth (2013) contain, among many others, various methods for the estimation of the variances of the test statistics or to get around it, also see Perron (2006) for a comprehensive review of the subject.

A host of problems arise in the context of change-point analysis for time series data, particularly those concerning the estimation of autocovariances (Tecuapetla-Gómez and Munk, 2017; Levine and Tecuapetla-Gomez, 2019) and the long run variance (Hall and Keilegom, 2003; Wu and Zhao, 2007; Khismatullina and Vogt, 2020; Chan, 2022; Bai and Wu, 2024). In particular, Tecuapetla-Gómez and Munk (2017) considers the estimation of autocovariances when the mean

function is piecewise smooth and the noise is $m$-dependent. Their methods rely on a high order difference of the data with difference gap $(m + 1)$. In this paper, we focus on a fundamental yet less discussed one: does the serial correlation exist when the data exhibit frequent mean shifts? We aim to address this question prior to change-point detection or autocorrelation estimation. In other words, we propose a test for the serial correlation that informs subsequent analysis based on its outcome. The validity of the proposed test is guaranteed under minimal assumptions, enhancing its utility as a preliminary diagnostic tool in data analysis. We also highlight that the proposed test is specifically designed for data with frequent change points, such as the nanopore sequencing data that motivated our research. As demonstrated in the numerical studies in Section 4.1.1, even a pseudo-oracle with full knowledge of the change-point locations fails to test for serial correlation if it first de-means the data segment-wise and then applies conventional white noise tests. This is not surprising: when the number of segments is large, piecewise centering introduces non-negligible artificial autocorrelations. In contrast, the proposed test effectively eliminates the impact of mean shifts, regardless of the number or magnitude of change points, ensuring a testing procedure that is faithful to its nominal level. As an additional contribution, we introduce a novel autocorrelation function (ACF) plot to visualize dependence patterns obscured by mean shifts.

This project is partially motivated by nanopore sequencing data analysis. Nanopore sequencing involves a nanoscale protein pore (nanopore) embedded in a membrane within an electrolytic solution. A consistent voltage generates an ionic current that propels negatively charged single-strand DNA or RNA sequences through the pore (Wang et al., 2021). As each base passes, it disrupts the current, which is indicative of its identity. The nanopore sequencing data record these temporal currents, often consisting of tens of thousands to millions of signal points. Change-point models are often used to model the piecewise constant mean structure of nanopore sequencing data. Basecalling algorithms then translate these data into nucleotide sequences, enabling genetic analyses such as gene expression studies and mutation detection. Early basecalling methods segment raw current data and use hidden Markov models (HMMs) to identify genetic bases, while recent approaches leverage deep learning models such as recurrent neural networks (RNNs). However, these methods do not explicitly incorporate autocovariance information, despite conjectures of positive autocorrelations in nanopore sequencing data (Garalde et al., 2012). Inferring autocorrelation structures is challenging due to frequent mean shifts caused by the rapid molecular transit through the nanopore (Fleming et al., 2021). Figure 1 illustrates examples of nanopore sequencing data with both conspicuous and subtle mean shifts, the latter being particularly difficult to detect. Developing methods to infer autocovariance structures without explicitly estimating the mean structure is critical for advancing nanopore sequencing data analysis.

Motivated by practical challenges and unmet methodological needs, we develop a novel statistical framework for testing and inferring covariance structures in nonstationary time series with mean shifts. Specifically, we consider all quadratic forms of the data that can be represented through symmetric Toeplitz matrices, which include the classic sample autocovariances for mean-zero time series as special cases. We then investigate whether certain quadratic statistics within this class can eliminate the impact of mean function when it is piecewise constant

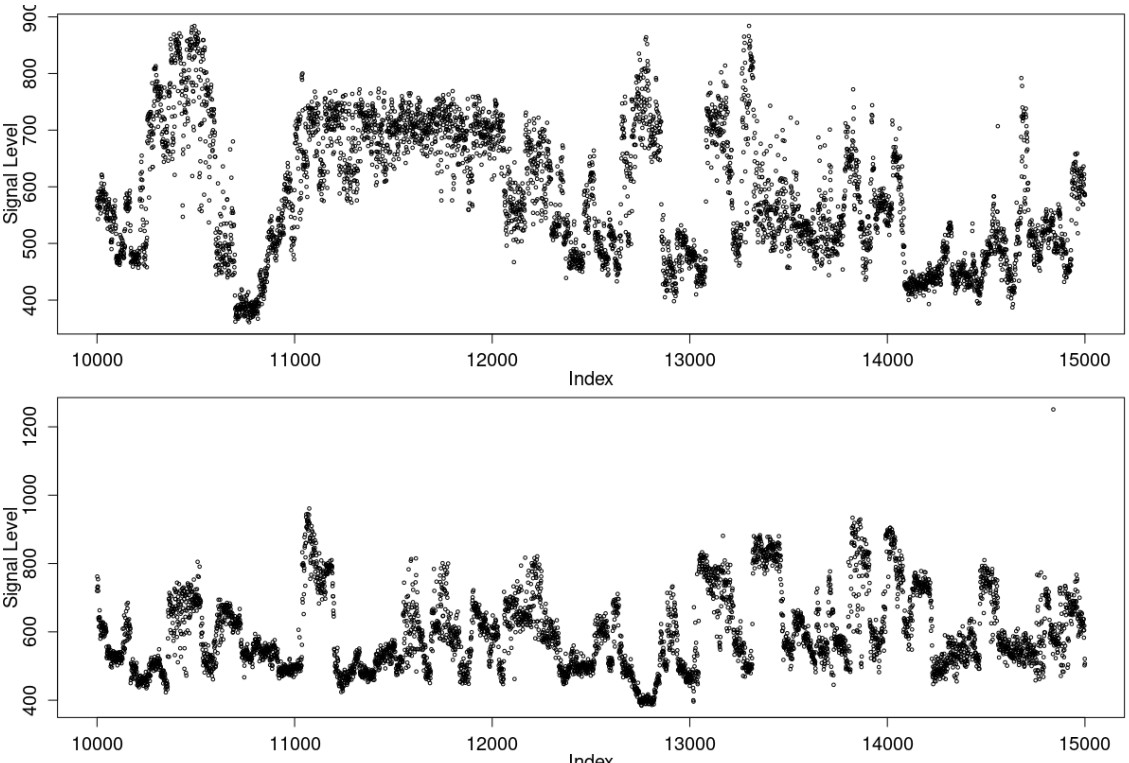

Figure 1: An illustration of nanopore sequencing data from Wang et al. (2024). Top: 5000 data points from sequence id=33; bottom: 5000 data points from sequence id=39. Each of these sequences contains numerous mean shifts.

and the additive noise is stationary. While classical sample autocovariances become misleading in the presence of mean shifts, we identify new quadratic forms that remain robust to such shifts. Section 2 illustrates how the influence of mean shifts can be removed in the expected values of certain types of quadratic forms and characterizes the class of such quadratic forms. In Section 3, we construct a Box-type portmanteau test based on these quadratic forms and provide related asymptotic theory. In Section 4, we demonstrate the effectiveness of the proposed methods using simulated and real data examples.

In summary, this work makes several contributions to the field of time series and change-point analysis. First, it tackles the challenging problem of inferring covariance structures in nonstationary time series with frequent and irregular mean shifts. We propose novel portmanteau tests that offer robust and practical tools for analyzing nonstationary time series where traditional methods are inadequate. Our techniques may also play an important role in change-point detection for time series data. Second, our visualization techniques provide an improved understanding of autocorrelation patterns in the presence of mean shifts, offering convenient tools for practitioners. Third, these methods are particularly useful for contemporary applications such as nanopore sequencing, where we have verified a conjecture by experts in nanopore sequencing data by showing significant positive autocorrelation in nanopore sequencing data. By addressing methodological gaps and introducing innovative tools, this work paves the way for more effective and precise analysis of nonstationary time series across a wide range of scientific disciplines.

## 2  Separation of the serial correlation and mean shifts

### 2.1  Model descriptions

Let $\{X_i\}_{i=1}^n$ be a sequence of random variables with piecewise constant means and additive stationary noises. Specifically, we assume

$$X_i = \theta_i + \varepsilon_i, \qquad 1 \leq i \leq n, \tag{1}$$

where the noise sequence $\{\varepsilon_i\}_{i=1}^n$ is mean-zero and stationary, and the mean parameters satisfy

$$\theta_1 = \theta_2 = \cdots = \theta_{\tau_1} \neq \theta_{\tau_1+1} = \cdots = \theta_{\tau_2} \neq \theta_{\tau_2+1} = \cdots \quad \cdots = \theta_{\tau_J} \neq \theta_{\tau_J+1} = \cdots = \theta_n. \tag{2}$$

Let $\boldsymbol{\tau} = (\tau_1, ..., \tau_J)^\top$ be the location vector of mean shifts, which partitions the entire sequence into $J+1$ segments with constant means. For theoretical derivations, it is convenient to represent the means of these segments as $\mu_1, \ldots, \mu_{J+1}$, respectively. Moreover, we denote the variance, lag-$h$ autocovariance and lag-$h$ autocorrelation of the noise sequence by $\gamma_0$, $\gamma_h$, and $\rho_h$, respectively.

The primary goal of this paper is to test whether $\rho_h = 0$ for all $h > 0$. In the absence of mean shifts, the process $\{X_i\}_{i=1}^n$ is stationary. The Box–Pierce test and its variants are fundamental tools for examining serial correlation. However, these methods perform poorly in the presence of mean shifts, as a non-constant mean structure can produce spurious autocorrelation, even if the underlying noises are independent. As demonstrated in our numerical studies, the type I error of the classical Box test is not well controlled when mean shifts are present.

In model (1), the data vector $\boldsymbol{X} = (X_1, X_2, \ldots, X_n)^\top$ is observed and indexed by the set of integers $[n] = \{1, ..., n\}$. We write (1) as $\boldsymbol{X} = \boldsymbol{\theta} + \boldsymbol{\varepsilon}$ using vector notations. For the mean vector $\boldsymbol{\theta}$, we denote by $\mathcal{L}(\boldsymbol{\theta})$ the minimal length of all constant segments in $\boldsymbol{\theta}$. That is, $\mathcal{L}(\boldsymbol{\theta}) = \min_{0 \leq j \leq J} \{\tau_{j+1} - \tau_j\}$ for $\boldsymbol{\theta}$ defined in (2) with the convention $\tau_0 = 0$ and $\tau_{J+1} = n$. In particular, $\mathcal{L}(\boldsymbol{\theta}) = n$ implies that all observations have the same mean; $\mathcal{L}(\boldsymbol{\theta}) = 1$ indicates that the mean may change at two consecutive positions. For any positive integer $L \leq n$, define the set of mean vectors

$$\Theta_L = \{\boldsymbol{\theta} \in \mathbb{R}^n : \mathcal{L}(\boldsymbol{\theta}) \geq L\}. \tag{3}$$

For instance, $n = 5$,

$$\Theta_2 = \{(c_1, c_1, c_1, c_1, c_1)^\top | c_1 \in \mathbb{R}\} \cup \{(c_2, c_2, c_3, c_3, c_3)^\top | c_2, c_3 \in \mathbb{R}\} \cup \{(c_4, c_4, c_4, c_5, c_5)^\top | c_4, c_5 \in \mathbb{R}\}.$$

$\Theta_L$ with varying $L$ defines a family of parameter spaces for the mean. A higher value of $L$ in (3) corresponds to a more restrictive model class. It is challenging to infer the autocorrelations for the model class $\Theta_L$ when $L$ is small due to frequent mean shifts.

We formalize our model description as follows.

**Condition 1.** $\{X_i\}_{i=1}^n$ follows model (1) with $\boldsymbol{\theta} = \mathrm{E}(\boldsymbol{X}) \in \Theta_L$.

**Condition 2.** The noise sequence $\{\varepsilon_i\}_{i=1}^n$ is mean-zero, strictly stationary, ergodic, and has finite fourth moments.

**Condition 3.** $\max_j \{(\mu_j - \mu_{j+1})^2\} = o(n)$.

Conditions 1 and 2 restate our model assumption on the mean and variance structures.

Condition 3 is mild and realistic. For example, it is reasonable to assume $\max_j\{(\mu_j - \mu_{j+1})^2\} = O(1)$ in many applications including nanopore studies. In fact, any jump of order $\log n$ or larger can be easily identified and removed; see Proposition 1 in Niu et al. (2016).

## 2.2 Quadratic forms

Classical Box type tests are built on the sample autocorrelations, or more fundamentally, sample autocovariances, which take the form of quadratic functions of the data. However, these standard tools are not directly applicable in our setting. To elaborate, we note that under model (1) and some mild additional conditions, the sample autocovariances $\check{\gamma}_h := \sum_{i=h+1}^{n}(X_i - \bar{X})(X_{i-h} - \bar{X})$ satisfy:

$$\check{\gamma}_h = \gamma_h + \check{\gamma}_h(\boldsymbol{\theta}) + O_p(n^{-1/2}), \quad h \geq 0;$$

where $\check{\gamma}_h(\boldsymbol{\theta}) := \sum_{i=h+1}^{n}(\theta_i - \bar{\theta})(\theta_{i-h} - \bar{\theta})$ is the "sample autocovariance" of the $\boldsymbol{\theta}$ sequence. Consequently, if the classical Box test is carried out based on the statistic $Q_m := n \sum_{h=1}^{m} \check{\rho}_m^2$, where $\check{\rho}_h := \check{\gamma}_h/\check{\gamma}_0$, then it holds that

$$Q_m = \sum_{h=1}^{m} \left( \frac{\gamma_h + \check{\gamma}_h(\boldsymbol{\theta})}{\gamma_0 + \check{\gamma}_0(\boldsymbol{\theta})} \right)^2 + O_p\left(n^{-1/2}\right). \tag{4}$$

This implies that when the data exhibit apparent autocorrelation, it is not immediately clear whether it reflects genuine temporal dependence in the noise sequence $\{\varepsilon_i\}$ or is merely an artifact of the piecewise constant mean sequence $\{\theta_i\}$. Furthermore, the effect of $\check{\gamma}_h(\theta)$ in (4) can be substantial even under the simplest change-point model. For example, if there is only one jump of size $2\delta$ at the center of the $\boldsymbol{\theta}$ sequence, it holds that $\lim_{n\to\infty} \check{\gamma}_h(\boldsymbol{\theta}) = \delta^2$ for any $h \geq 0$. Therefore, to test for serial correlation, it is essential to isolate the temporal dependence of the $\boldsymbol{\varepsilon}$ sequence by filtering out the effect of the mean function.

All the sample autocovariances are quadratic forms of the data sequence, and more importantly, information about autocovariance is inherently embedded in all pairwise products $X_i X_j$. Motivated by this, we consider general quadratic forms of the type $\boldsymbol{X}^\top \boldsymbol{A} \boldsymbol{X}$ where $\boldsymbol{A}$ is a symmetric Toeplitz matrix of the form

$$\begin{pmatrix} a_0 & a_1 & a_2 & \cdots & a_{n-1} \\ a_1 & a_0 & a_1 & \cdots & a_{n-2} \\ a_2 & a_1 & a_0 & \cdots & a_{n-3} \\ \vdots & \vdots & \vdots & \ddots & \vdots \\ a_{n-1} & a_{n-2} & a_{n-3} & \cdots & a_0 \end{pmatrix}, \tag{5}$$

and look for suitable choices of $\boldsymbol{A}$ as alternatives to the standard sample autocovariances, with the goal of mitigating the influence of the mean vector $\boldsymbol{\theta}$. We start with a simple observation.

**Proposition 1** *Let $\boldsymbol{A}$ be a symmetric Toeplitz matrix of the form* (5). *For $\boldsymbol{X}$ generated from model* (1) *where the noise is mean zero and stationary with a finite variance, we have*

$$\mathrm{E}\left(\boldsymbol{X}^\top \boldsymbol{A} \boldsymbol{X}\right) = \sum_{h=1-n}^{n-1}(n - |h|)a_{|h|}\gamma_h + \boldsymbol{\theta}^\top \boldsymbol{A} \boldsymbol{\theta}. \tag{6}$$

Removing the impact of the mean function amounts to ensuring that the second term $\boldsymbol{\theta}^\top \boldsymbol{A}\boldsymbol{\theta} = 0$ in (6). However, unless $\boldsymbol{A}$ is the zero matrix, this condition cannot hold for all $\boldsymbol{\theta} \in \mathbb{R}^n$. Remarkably, the piecewise constant structure of the mean function in model (1) allows for a class of quadratic forms whose expectations are unaffected by $\boldsymbol{\theta}$, as established by the following theorem.

**Theorem 1** *Let $\boldsymbol{A}$ be a symmetric Toeplitz matrix of the form* (5)*, and $L$ be an integer with $1 \leq L < n/2$. We conclude that $\boldsymbol{\theta}^\top \boldsymbol{A}\boldsymbol{\theta} = 0$ for all $\boldsymbol{\theta} \in \Theta_L$ if and only if the following equations hold.*

$$a_0 + 2a_1 + \cdots + 2a_L = 0, \tag{7}$$

$$a_1 + 2a_2 + \cdots + La_L = 0, \tag{8}$$

$$a_{L+1} = a_{L+2} = \cdots = a_{n-L-1} = 0, \tag{9}$$

$$La_{n-L} + (L-1)a_{n-L+1} + \cdots + a_{n-1} = 0. \tag{10}$$

Within the class of quadratic forms characterized by Theorem 1, we further consider a subclass that is invariant under a global mean shift. Specifically, we seek matrices $\boldsymbol{A}$ such that $\boldsymbol{X}^\top \boldsymbol{A}\boldsymbol{X} = (\boldsymbol{X}+c\boldsymbol{1})^\top \boldsymbol{A}(\boldsymbol{X}+c\boldsymbol{1})$ for all $c \in \mathbb{R}$. This invariance property is automatically satisfied by the sample autocovariances when computed after centering the data by the sample mean. However, it requires a nontrivial calibration if the quadratic forms satisfying Theorem 1 are concerned.

**Proposition 2** *Suppose $L$ is an integer with $1 \leq L < n/2$. Let $\boldsymbol{A}$ be a symmetric Toeplitz matrix of the form* (5) *satisfying the equations* (7)–(10) *in Theorem 1. The quadratic form $\boldsymbol{X}^\top \boldsymbol{A}\boldsymbol{X}$ is invariant under a global mean shift if and only if the following additional equations hold,*

$$a_i = a_{n-i}, \quad i = 1, \ldots, L, \tag{11}$$

*or equivalently, $\boldsymbol{A}$ is a circulant matrix of the form*

$$
\begin{pmatrix}
a_0 & a_1 & a_2 & \cdots & a_{L-1} & a_L & 0 & \cdots & 0 & a_L & a_{L-1} & a_{L-2} & \cdots & a_2 & a_1 \\
a_1 & a_0 & a_1 & \cdots & a_{L-2} & a_{L-1} & a_L & \cdots & 0 & 0 & a_L & a_{L-1} & \cdots & a_3 & a_2 \\
a_2 & a_1 & a_0 & \cdots & a_{L-3} & a_{L-2} & a_{L-1} & \cdots & 0 & 0 & 0 & a_L & \cdots & a_4 & a_3 \\
\vdots & \vdots & \vdots & \vdots & \vdots & \vdots & \vdots & \vdots & \vdots & \vdots & \vdots & \vdots & \vdots & \vdots & \vdots \\
a_1 & a_2 & a_3 & \cdots & a_L & 0 & 0 & \cdots & a_L & a_{L-1} & a_{L-2} & a_{L-3} & \cdots & a_1 & a_0
\end{pmatrix}. \tag{12}
$$

To sum up, in order for the quadratic form $\boldsymbol{X}^\top \boldsymbol{A}\boldsymbol{X}$ to be both independent of $\boldsymbol{\theta}$ in expectation when $\boldsymbol{\theta} \in \Theta_L$ and invariant under a global mean shift, $\boldsymbol{A}$ must be a circulant matrix with the first row $(a_0, a_1, \ldots, a_L, 0, \cdots, 0, a_L, \cdots, a_2, a_1)$, where the coefficients $a_0, a_1, \ldots, a_L$ satisfy (7) and (8). We denote the set of all such $\boldsymbol{A}$ matrices by $\mathcal{A}_L$. When $\boldsymbol{A} \in \mathcal{A}_L$, the quadratic form $\boldsymbol{X}^\top \boldsymbol{A}\boldsymbol{X}$ has a more explicit representation.

**Proposition 3** *For $\boldsymbol{A} \in \mathcal{A}_L$ of the form in (12), we have*

$$\boldsymbol{X}^\top \boldsymbol{A} \boldsymbol{X} = -\sum_{h=1}^{L} a_h T_h, \tag{13}$$

*where $T_h = \sum_{i=1}^n (X_i - X_{i+h})^2$, $h = 1, \ldots, L$.*

Our approach is to base the test for serial correlation on the quadratic forms, so it is also necessary to remove the variance $\gamma_0$ from (6).

**Proposition 4** *Assume Conditions 1 and 2. Suppose $\boldsymbol{A} \in \mathcal{A}_L$, the expectation of $\boldsymbol{X}^\top \boldsymbol{A} \boldsymbol{X}$ does not depend on $\gamma_0$ if and only if $a_0 = 0$, which is equivalent to $a_1 + \cdots + a_L = 0$.*

Therefore, we have narrowed the choice of the quadratic forms down to the class $\mathcal{A}_L^\circ$, represented by vectors in $\mathbb{R}^L$ as follows.

$$\mathcal{A}_L^\circ = \{(a_1, \ldots, a_L)^\top \in \mathbb{R}^L \mid a_1 + \cdots + a_L = 0, \;\; a_1 + 2a_2 + \cdots + La_L = 0\}. \tag{14}$$

By this convention, there is a natural embedding $\mathcal{A}_K^\circ \subset \mathcal{A}_L^\circ$ when $K < L$, which maps $(a_1, \ldots, a_K)^\top$ to $(a_1, \ldots, a_K, 0, \ldots, 0)^\top$. In other words, $\{\mathcal{A}_L^\circ\}$ represents a nested family of quadratic forms. For any $\boldsymbol{a} = (a_1, \ldots, a_L)^\top \in \mathcal{A}_L^\circ$, we use $\boldsymbol{A_a}$ to denote the corresponding circulant matrix starting from the row $(0, a_1, \ldots, a_L, 0, \cdots, 0, a_L, \cdots, a_2, a_1)$, and $\boldsymbol{X}^\top \boldsymbol{A_a} \boldsymbol{X}$ to denote the corresponding quadratic forms.

## 3 Shift-Immune Portmanteau Test

### 3.1 Construction of the Test Statistic

In designing the test, we are primarily motivated by data that may exhibit frequent changes in the mean function $\boldsymbol{\theta}$. The class of quadratic forms discussed in the previous section captures information about the autocovariances while eliminating the influence of $\boldsymbol{\theta}$, thus serving as the building blocks of the proposed test statistic. Before proceeding further, we remark that the discussion in Section 2.2 relies on the key quantity $L$, the minimal segment length of $\boldsymbol{\theta}$. Our strategy is to construct the test statistic based on the class $\mathcal{A}_{m+2}^\circ \subset \mathcal{A}_L^\circ$ with the assumption $m + 2 \leq L$ or $m + 2 \leq L/2$, where $m$ is a pre-specified number of lags, as in all Box-type tests. Since traditional Box-type tests typically involve a modest number of autocorrelations, we do not aim to use a large $m$ either. We believe that $m + 2 \leq L$ or $m + 2 \leq L/2$ is a reasonable assumption and do not require any additional conditions on $L$.

With a specified number of lags $m$, we focus on the quadratic forms parametrized by the linear space $\mathcal{A}_{m+2}^\circ$. As shown in (6), if $\boldsymbol{\theta} \in \Theta_L$, then for any $\boldsymbol{a} \in \mathcal{A}_{m+2}^\circ \subset \mathcal{A}_L^\circ$,

$$\mathrm{E}\left(\boldsymbol{X}^\top \boldsymbol{A_a} \boldsymbol{X}\right) = \sum_{h=-(m+2)}^{m+2} (n - |h|) a_{|h|} \gamma_h + \sum_{h=-(m+2)}^{m+2} |h| a_{|h|} \gamma_{n-|h|}.$$

The expectation involves two sets of autocovariances $\{\gamma_1, \ldots, \gamma_{m+2}\}$ and $\{\gamma_{n-1}, \gamma_{n-2}, \ldots, \gamma_{n-m-2}\}$. Under any short-range dependence condition, the second set becomes negligible when the sam-

ple size grows, so we will essentially use the quadratic form to test whether the first set of autocovariances are zero or not.

It is evident that $\mathcal{A}_{m+2}^\circ$ is a $m$-dimensional linear subspace of $\mathbb{R}^{m+2}$. To accumulate the information on $\{\gamma_1, \ldots, \gamma_{m+2}\}$, we pick a set of linearly independent elements $\{\boldsymbol{a}_h, \, h = 1, 2, \ldots, m\}$ from $\mathcal{A}_{m+2}^\circ$, and consider the corresponding quadratic forms $\{\boldsymbol{X}^\top \boldsymbol{A}_{\boldsymbol{a}_h} \boldsymbol{X}\}$. Define an $(m+2) \times m$ matrix $\boldsymbol{C} = [\boldsymbol{a}_1, \boldsymbol{a}_2, \ldots, \boldsymbol{a}_m]$ and an $(m+2)$-dimensional vector $\boldsymbol{T}_{m+2} = (T_1, \ldots, T_{m+2})^\top$. In view of (13) in Proposition 3, these quadratic forms constitute the random vector $\boldsymbol{C}^\top \boldsymbol{T}_{m+2}$. Since the expectation of this random vector involves linear combinations of $\{\gamma_1, \ldots, \gamma_{m+2}\}$, it is natural to "square" $\boldsymbol{C}^\top \boldsymbol{T}_{m+2}$ to form the test statistic. As will be elaborated in Lemma 1, the random vector $\boldsymbol{T}_{m+2}$ has nonzero covariances induced by the mean structure $\boldsymbol{\theta}$, even under the null hypothesis that the $\varepsilon_i$'s are independent and identically distributed (IID). Therefore, the random vector $\boldsymbol{C}^\top \boldsymbol{T}_{m+2}$ must be properly scaled with respect to its covariance matrix when forming the test statistic.

Let $\boldsymbol{\Sigma}$ be any nonsingular $(m+2) \times (m+2)$ covariance matrix. If $\boldsymbol{\Sigma}$ is the covariance matrix of $\boldsymbol{T}_{m+2}$, then the covariance matrix of $\boldsymbol{C}^\top \boldsymbol{T}_{m+2}$ is given by $\boldsymbol{C}^\top \boldsymbol{\Sigma} \boldsymbol{C}$. The test statistic that we will propose takes the form:

$$\boldsymbol{T}_{m+2}^\top \boldsymbol{C} \left[ \boldsymbol{C}^\top \boldsymbol{\Sigma} \boldsymbol{C} \right]^{-1} \boldsymbol{C}^\top \boldsymbol{T}_{m+2}. \tag{15}$$

A more detailed discussion on the calculation and estimation of the test statistic is deferred to Section 3.4. Here, we simply note that the test statistic (15) does not depend on any specific choice of the basis $\boldsymbol{a}_h$ of $\mathcal{A}_{m+2}^\circ$ or the matrix $\boldsymbol{C}$. Suppose $\boldsymbol{U}$ is a $(m+2) \times m$ matrix whose columns form another basis of $\mathcal{A}_{m+2}^\circ$, then we can always write $\boldsymbol{C}$ as $\boldsymbol{U}\boldsymbol{Q}$ where $\boldsymbol{Q}$ is a nonsingular square matrix. Replacing $\boldsymbol{C}$ by $\boldsymbol{U}\boldsymbol{Q}$, the preceding equation becomes

$$\boldsymbol{T}_{m+2}^\top \boldsymbol{U} \boldsymbol{Q} \left[ \boldsymbol{Q}^\top \boldsymbol{U}^\top \boldsymbol{\Sigma} \boldsymbol{U} \boldsymbol{Q} \right]^{-1} \boldsymbol{Q}^\top \boldsymbol{U}^\top \boldsymbol{T}_{m+2} = \boldsymbol{T}_{m+2}^\top \boldsymbol{U} \left[ \boldsymbol{U}^\top \boldsymbol{\Sigma} \boldsymbol{U} \right]^{-1} \boldsymbol{U}^\top \boldsymbol{T}_{m+2}. \tag{16}$$

Therefore, the quantity in (15) does not depend on any specific choice of $\boldsymbol{C}$ as long as its column space spans $\mathcal{A}_{m+2}^\circ$. In other words, our proposed test statistic is unique for each $m$.

## 3.2 Choice of Basis

The aforementioned invariance property with respect to $\boldsymbol{C}$ allows us to choose any suitable $\boldsymbol{C}$ to facilitate the subsequent discussion. For the sake of interpretability, we make the following choice of $\boldsymbol{a}_h = (a_{h1}, a_{h2}, \ldots, a_{h,m+2})^\top$, $h = 1, 2, \ldots, m$:

$$\begin{aligned} a_{hh} &= \frac{1}{2n}, & a_{h,m+1} &= -\frac{m+2-h}{2n} \\ a_{h,m+2} &= \frac{m+1-h}{2n}, & a_{hi} &= 0 \text{ for } i \notin \{h, \, m+1, \, m+2\}. \end{aligned} \tag{17}$$

Such a choice can be motivated by presuming an additional $m$-dependence structure of the noise sequence $\{\varepsilon_i\}$ on top of Condition 2. We emphasize that this $m$-dependence assumption is introduced solely for the interpretation of the construction in (17); it is not required for the validity of our test procedure. For each $1 \leq h \leq m$, let $\boldsymbol{A}_h$ be the circulant matrix induced by

$\boldsymbol{a}_h$. We begin by explicitly expanding the quadratic form $\boldsymbol{X}^\top \boldsymbol{A}_h \boldsymbol{X}$ and assigning it the name

$$
\begin{aligned}
\hat{\gamma}_h &:= \boldsymbol{X}^\top \boldsymbol{A}_h \boldsymbol{X} \\
&= \frac{1}{n} \left( \sum_{i=1}^n X_i X_{i+h} - (m+2-h) \sum_{i=1}^n X_i X_{i+m+1} + (m+1-h) \sum_{i=1}^n X_i X_{i+m+2} \right) \quad (18) \\
&= (2n)^{-1} \left[ -T_h + (m+2-h)T_{m+1} - (m+1-h)T_{m+2} \right],
\end{aligned}
$$

where we use the convention that $X_{n+i} = X_i$ whenever the subscript goes beyond $n$. We further define:

$$
\hat{\gamma}_0 = (2n)^{-1} \left[ (m+2)T_{m+1} - (m+1)T_{m+2} \right]. \quad (19)
$$

**Proposition 5** *Under Conditions 1 and 2, if $m + 2 \leq L$, $m + 2 < n/2$, and that $\{\varepsilon_i\}$ is $m$-dependent, then*

$$
\begin{aligned}
\mathrm{E}\hat{\gamma}_0 &= \gamma_0, \\
\mathrm{E}\hat{\gamma}_h &= (1 - h/n)\gamma_h, \qquad\qquad\qquad\qquad 1 \leq h \leq m.
\end{aligned}
$$

The proposition confirms that $\{\hat{\gamma}_h\}$ are valid estimates of the corresponding autocovariances under the $m$-dependence condition, thus justifying their appellations. More importantly, they are all valid under the presence of any mean function $\boldsymbol{\theta} \in \Theta_L$, and are invariant under the global mean shift. Consequently, we define

$$
\hat{\rho}_h = \hat{\gamma}_h / \hat{\gamma}_0, \quad h \geq 1. \quad (20)
$$

### 3.3 What is being tested?

In this subsection, we investigate when the test based on (15) and the equivalent form (16) has power asymptotically. In view of (14), $\mathcal{A}_{m+2}^\circ$ is naturally a linear subspace of $\mathbb{R}^{m+2}$. Let $\mathcal{P}_{m+2}$ denote the orthogonal projection matrix onto the subspace $\mathcal{A}_{m+2}^\circ$.

**Proposition 6** *Under Conditions 1, 2, and 3, if $m$ is an integer such that $m + 2 \leq L$, and $m + 2 < n/2$, then*

$$
\mathcal{P}_{m+2} \boldsymbol{T}_{m+2} / (2n) \xrightarrow{P} \mathcal{P}_{m+2} \boldsymbol{\gamma}_{m+2}, \quad (21)
$$

*where $\boldsymbol{\gamma}_{m+2} = (\gamma_1, \gamma_2, \ldots, \gamma_{m+2})^\top$.*

Consequently, the test statistic in (15) essentially tests whether the projection $\mathcal{P}_{m+2}\boldsymbol{\gamma}_{m+2}$ is zero. Indeed, (15) can be rewritten as

$$
\boldsymbol{T}_{m+2}^\top \boldsymbol{C} \left[ \boldsymbol{C}^\top \boldsymbol{\Sigma} \boldsymbol{C} \right]^{-1} \boldsymbol{C}^\top \boldsymbol{T}_{m+2} = (\mathcal{P}_{m+2}\boldsymbol{T}_{m+2})^\top \boldsymbol{C} \left[ \boldsymbol{C}^\top \boldsymbol{\Sigma} \boldsymbol{C} \right]^{-1} \boldsymbol{C}^\top (\mathcal{P}_{m+2}\boldsymbol{T}_{m+2}),
$$

since $\mathcal{P}_{m+2}\boldsymbol{C} = \boldsymbol{C}$.

Our original objective is to assess the white-noise hypothesis, namely that the entire autocorrelation sequence vanishes. However, according to the discussion above, the proposed test is effectively testing if $\mathcal{P}_{m+2}\boldsymbol{\gamma}_{m+2} = \boldsymbol{0}$. Thus, as with the classical Box-type tests, there is an unavoidable gap between the full population-level target and the finite-dimensional hypothesis actually being tested.

Furthermore, the test would have no power if $\mathcal{P}_{m+2}\boldsymbol{\gamma}_{m+2} = \mathbf{0}$. This is the price we pay for eliminating the influence of the nontrivial piecewise-constant mean vector $\boldsymbol{\theta}$, and it merits further discussion. In classical time series analysis, especially in diagnostics, when checking whether a series has autocorrelations, it has been customary to apply Box-type tests for multiple values of $m$, as has been rendered in some standard R packages, e.g. the `tsdiag()` function in the base `stats` package. If $\mathcal{P}_{m+2}\boldsymbol{\gamma}_{m+2} = \mathbf{0}$ for every $m$, then it entails that, in view of (17), $\gamma_m - 2\gamma_{m+1} + \gamma_{m+2} = 0$ for every $m \geq 1$, or equivalently,

$$\gamma_{m+2} - \gamma_{m+1} = \gamma_{m+1} - \gamma_m, \text{ for all } m \geq 1.$$

The general solution of this difference equation takes the form $\gamma_h = c_0 + c_1 h$, where $c_0$ and $c_1$ are constants. However, such a linear form for the autocovariance sequence $\gamma_h$ is incompatible with the short-range dependence condition $\gamma_h \to 0$ as $h \to \infty$. Therefore, under this condition, the test must have asymptotic power for some $m$, provided that not all $\gamma_h$ are zero. We thus conclude that the power loss induced by removing the impact of $\boldsymbol{\theta}$ is limited. Thus, except for pathological long-range linear autocovariance structures incompatible with short-range dependence, the proposed test retains asymptotic power.

### 3.4 Asymptotic theory

Before stating the asymptotic results, we introduce some notations. Let $\kappa_4 = \mathrm{E}(\varepsilon_1^4)/\gamma_0^2$. Define $w = W(\boldsymbol{\theta})/(n\gamma_0)$, where

$$W(\boldsymbol{\theta}) = \sum_{i=1}^{n}(\theta_i - \theta_{i+1})^2 = \sum_{j=1}^{J+1}(\mu_j - \mu_{j+1})^2.$$

Let $\boldsymbol{I}_m$ be the $m \times m$ identity matrix, $\boldsymbol{H}_m = (H_{ij})$ be an $m \times m$ matrix with $H_{ij} = \min\{i, j\}$, $\boldsymbol{\eta}_m$ be the vector $(1, 2, \ldots, m)^\top$, $\mathbf{0}_m$ and $\mathbf{1}_m$ be vectors of length $m$ with all entries equal to 0 and 1, respectively. In the asymptotic analysis, we treat $\boldsymbol{\theta}$ and its characteristics such as $J$, $\boldsymbol{\mu}$, and $w$ as functions of $n$.

As discussed in Section 3.1, the test statistic (15) does not depend on the particular choice of $\boldsymbol{C}$, as long as its column space is the same as $\mathcal{A}_{m+2}^{\circ}$. Therefore, we adopt the version of $\boldsymbol{C}$ described in (17) in Section 3.2, and use the notations $\hat{\gamma}_h$ and $\hat{\rho}_h$ defined in (18) and (20), which facilitate the formulation of the results. We begin with a central limit theorem for $\hat{\boldsymbol{\gamma}}_m = (\hat{\gamma}_1, \ldots, \hat{\gamma}_m)^\top$. Since the variance of $\hat{\boldsymbol{\gamma}}_m$ depends on $w$, which is allowed to vary as $n$ grows, we need to normalize $\hat{\boldsymbol{\gamma}}_m$ by its asymptotic covariance matrix (scaled by $1/n$)

$$\begin{aligned}
\boldsymbol{\Sigma}_{\gamma,w} :=& \gamma_0^2 \{ \boldsymbol{I}_m + \left[ (2m^2 + 6m + 5) + 2(m^2 + 3m + 2)w \right] \mathbf{1}_m \mathbf{1}_m^\top \\
& - \left[ (2m+3) + 2(m+2)w \right] (\boldsymbol{\eta}_m \mathbf{1}_m^\top + \mathbf{1}_m \boldsymbol{\eta}_m^\top) \\
& + (2 + 2w)\boldsymbol{\eta}_m \boldsymbol{\eta}_m^\top + 2w\boldsymbol{H}_m \}.
\end{aligned} \tag{22}$$

We append the subscript $w$ to $\boldsymbol{\Sigma}_{\gamma,w}$ to emphasize its dependence on $w$.

**Theorem 2** *Under conditions 1, 2, and 3, and the null hypothesis that $\varepsilon_1, \ldots, \varepsilon_n$ are IID,*

*suppose $m$ is an integer such that $m + 2 \leq L/2$, and $m + 2 < n/2$, then $\hat{\gamma}_0$ and $\hat{\gamma}$ satisfy*

$$\hat{\gamma}_0 \xrightarrow{P} \gamma_0, \quad \boldsymbol{\Sigma}_{\gamma,w}^{-1/2}(\sqrt{n}\hat{\gamma}_m) \xrightarrow{D} \mathcal{N}(\boldsymbol{0}_m, \boldsymbol{I}_m), \quad as \ n \to \infty.$$

*Moreover, under a relaxed condition $m + 2 \leq L$, it holds that*

$$\lim_{n \to \infty} \boldsymbol{\Sigma}_{\gamma,2w}^{-1/2}(\sqrt{n}\hat{\gamma}_m) \preceq \mathcal{N}(\boldsymbol{0}_m, \boldsymbol{I}_m).$$

The notation $\preceq$ in the second statement of Theorem 2 indicates stochastic dominance. Specifically, for a sequence of random vectors $\{\boldsymbol{Q}_n\}$ and a "limit" random vector $\boldsymbol{Q}$, the statement $\lim_{n \to \infty} \boldsymbol{Q}_n \preceq \boldsymbol{Q}$ means that

$$\overline{\lim_{n \to \infty}} \ P\left[\left|\boldsymbol{b}^\top \boldsymbol{Q_n}\right| \geq x\right] \leq P\left[\left|\boldsymbol{b}^\top \boldsymbol{Q}\right| \geq x\right],$$

for any unit vector $\boldsymbol{b}$ and any positive number $x$.

Let $\hat{\boldsymbol{\rho}}_m = (\hat{\rho}_1, \ldots, \hat{\rho}_m)^\top = \hat{\gamma}_m/\hat{\gamma}_0$. It follows Theorem 2 and Slutsky's theorem that $\boldsymbol{\Sigma}_{\rho,w}^{-1/2}(\sqrt{n}\hat{\boldsymbol{\rho}}) \to \mathcal{N}(\boldsymbol{0}_m, \boldsymbol{I}_m)$, where $\boldsymbol{\Sigma}_{\rho,w} = \boldsymbol{\Sigma}_{\gamma,w}/\gamma_0^2$, and $\boldsymbol{\Sigma}_{\gamma,w}$ is defined in (22). In fact, this asymptotic property holds for any autocorrelation estimators $\hat{\boldsymbol{\gamma}}/\tilde{\gamma}_0$ whenever $\tilde{\gamma}_0$ is consistent. Moreover, the asymptotic covariance $\boldsymbol{\Sigma}_{\rho,w}$ depends on only one unknown quantity $w$, which makes it straightforward to estimate. For example, we could employ

$$\hat{w}_1 = 2(n\hat{\gamma}_0)^{-1}\left(\sum_{i=1}^n X_i X_{i+m+2} - \sum_{i=1}^n X_i X_{i+m+1}\right) = (n\hat{\gamma}_0)^{-1}(T_{m+2} - T_{m+1}). \tag{23}$$

In addition, under the null hypothesis, $w$ can be estimated from a regression model studied in Hao et al. (2023)

$$(2n)^{-1}T_h = \alpha + h\beta + e_h, \qquad h = 1, \ldots, m+2,$$

where $(\alpha, \beta)^\top = (\gamma_0, w\gamma_0/2)^\top$. The least squares estimator is defined as

$$(\hat{\alpha}, \hat{\beta})^\top = (\boldsymbol{Z}_{m+2}^\top \boldsymbol{Z}_{m+2})^{-1}\boldsymbol{Z}_{m+2}^\top \boldsymbol{T}_{m+2}/2n, \tag{24}$$

where $\boldsymbol{Z}_{m+2} = (\boldsymbol{1}_{m+2}, \boldsymbol{\eta}_{m+2})$. Consequently, $w$ can be estimated by

$$\hat{w}_2 = 2\hat{\beta}/\hat{\alpha}.$$

The empirical version of $\boldsymbol{\Sigma}_{\rho,w}$ is defined by plugging $\hat{w}$ ($\hat{w}_1$ or $\hat{w}_2$) in $\boldsymbol{\Sigma}_{\rho,w} = \boldsymbol{\Sigma}_{\gamma,w}/\gamma_0^2$, i.e.,

$$\begin{aligned}
\boldsymbol{\Sigma}_{\rho,\hat{w}} = {}&\boldsymbol{I}_m + \left[(2m^2 + 6m + 5) + 2(m^2 + 3m + 2)\hat{w}\right]\boldsymbol{1}_m\boldsymbol{1}_m^\top \\
&- \left[(2m + 3) + 2(m + 2)\hat{w}\right](\boldsymbol{\eta}_m\boldsymbol{1}_m^\top + \boldsymbol{1}_m\boldsymbol{\eta}_m^\top) \\
&+ (2 + 2\hat{w})\boldsymbol{\eta}_m\boldsymbol{\eta}_m^\top + 2\hat{w}\boldsymbol{H}_m.
\end{aligned} \tag{25}$$

**Theorem 3** *Under conditions 1, 2, and 3, and the null hypothesis that $\varepsilon_1, \ldots, \varepsilon_n$ are IID, if*

$m + 2 \leq L/2$, and $m + 2 < n/2$, then

$$n\hat{\boldsymbol{\rho}}^\top \boldsymbol{\Sigma}_{\rho,\hat{w}}^{-1} \hat{\boldsymbol{\rho}} \xrightarrow{D} \chi_m^2, \tag{26}$$

where $\boldsymbol{\Sigma}_{\rho,\hat{w}}$ is defined in (25), $\hat{\boldsymbol{\rho}} = \hat{\boldsymbol{\gamma}}/\hat{\gamma}_0$ for any consistent estimator $\hat{\gamma}_0$.

Moreover, under a relaxed condition $m + 2 \leq L$, it holds that

$$\lim_{n\to\infty} n\hat{\boldsymbol{\rho}}^\top \boldsymbol{\Sigma}_{\rho,2\hat{w}}^{-1} \hat{\boldsymbol{\rho}} \preceq \chi_m^2.$$

We next discuss the asymptotic power of the proposed test. The test statistic $n\hat{\boldsymbol{\rho}}^\top \boldsymbol{\Sigma}_{\rho,\hat{w}}^{-1} \hat{\boldsymbol{\rho}}$ depends on $\hat{\boldsymbol{\gamma}}_m$, $\hat{\gamma}_0$ and $\hat{w}$. We have mentioned two estimates $\hat{w}_1$ and $\hat{w}_2$ of $w$, relying, respectively, on two estimates for $\gamma_0$: $\hat{\gamma}_0$ in (19) and the EVE estimator $\hat{\alpha}$ in (24). Being consistent under the null hypothesis, these estimates become biased under the alternative. Specifically, direct calculations show that

$$\bar{\mathrm{E}}\hat{\gamma}_0 := \lim_{n\to\infty} \mathrm{E}\hat{\gamma}_0 = \gamma_0 - (m+2)\gamma_{m+1} + (m+1)\gamma_{m+2},$$

$$\bar{\mathrm{E}}\hat{\alpha} := \lim_{n\to\infty} \mathrm{E}\hat{\alpha} = \gamma_0 - [1,0](\boldsymbol{Z}_{m+2}^\top \boldsymbol{Z}_{m+2})^{-1}\boldsymbol{Z}_{m+2}^\top \boldsymbol{\gamma}_{m+2},$$

$$\bar{\mathrm{E}}(\hat{\gamma}_0 \hat{w}_1) := \lim_{n\to\infty} \mathrm{E}(\hat{\gamma}_0 \hat{w}_1) = \gamma_0 w + 2\gamma_{m+1} - 2\gamma_{m+2},$$

$$\bar{\mathrm{E}}(\hat{\alpha}\hat{w}_2) := \lim_{n\to\infty} \mathrm{E}(\hat{\alpha}\hat{w}_2) = \gamma_0 w - [2,0](\boldsymbol{Z}_{m+2}^\top \boldsymbol{Z}_{m+2})^{-1}\boldsymbol{Z}_{m+2}^\top \boldsymbol{\gamma}_{m+2}.$$

For the power analysis, we need to impose conditions on these expectations, i.e. what these estimates are actually estimating. In the following proposition, with a slight abuse of notation, the pair $(\hat{\gamma}_0, \hat{w})$ could be either the pair $(\hat{\gamma}_0, \hat{w}_1)$ or the pair $(\hat{\alpha}, \hat{w}_2)$.

**Proposition 7** *Assume conditions 1, 2 and 3. Also, assume* $\bar{\mathrm{E}}\hat{\gamma}_0 > 0$, *and the matrix*

$$\begin{aligned}
\bar{\Sigma} :=& (\bar{\mathrm{E}}\hat{\gamma}_0)\boldsymbol{I}_m + \left[(2m^2 + 6m + 5)(\bar{\mathrm{E}}\hat{\gamma}_0) + 2(m^2 + 3m + 2)\bar{\mathrm{E}}(\hat{\gamma}_0\hat{w})\right] \boldsymbol{1}_m\boldsymbol{1}_m^\top \\
& - \left[(2m+3)(\bar{\mathrm{E}}\hat{\gamma}_0) + 2(m+2)\bar{\mathrm{E}}(\hat{\gamma}_0\hat{w})\right] (\boldsymbol{\eta}_m\boldsymbol{1}_m^\top + \boldsymbol{1}_m\boldsymbol{\eta}_m^\top) \\
& + 2[(\bar{\mathrm{E}}\hat{\gamma}_0) + \bar{\mathrm{E}}(\hat{\gamma}_0\hat{w})]\boldsymbol{\eta}_m\boldsymbol{\eta}_m^\top + 2\bar{\mathrm{E}}(\hat{\gamma}_0\hat{w})\boldsymbol{H}_m
\end{aligned}$$

*is positive definite. If* $m + 2 \leq L/2$, $m + 2 < n/2$, *and* $\mathcal{P}_{m+2}\boldsymbol{\gamma}_{m+2} \neq \boldsymbol{0}$, *then*

$$\lim_{n\to\infty} P\left[n\hat{\boldsymbol{\rho}}^\top \boldsymbol{\Sigma}_{\rho,\hat{w}}^{-1} \hat{\boldsymbol{\rho}} > c\right] = 1$$

*for any constant* $c \geq 0$.

### 3.5 Test procedures

In light of Theorem 3, we propose using $n\hat{\boldsymbol{\rho}}_m^\top \boldsymbol{\Sigma}_{\rho,\hat{w}}^{-1} \hat{\boldsymbol{\rho}}_m$ as a test statistic. It is clear that the proposed test statistic bears strong resemblance to the classical Box-type test (Box and Pierce, 1970), with the important difference that the quadratic norm of $\hat{\boldsymbol{\rho}}_m$ has to be calculated under a proper precision matrix $\boldsymbol{\Sigma}_{\rho,\hat{w}}^{-1}$. The construction of the test statistic, including the calculation of $\hat{\rho}_m$ and its covariance matrix, is motivated and necessitated by data with frequent mean shifts. We therefore refer to $n\hat{\boldsymbol{\rho}}_m^\top \boldsymbol{\Sigma}_{\rho,\hat{w}}^{-1} \hat{\boldsymbol{\rho}}_m$ as the Shift-Immune Portmanteau (SIP) test statistic. Based

on the theoretical results, we introduce two versions of the SIP test below.

Similar to the Box tests, we need to pre-specify the lag $m$ to construct the test statistic. Given $m$, we estimate the vector of $\hat{\boldsymbol{\gamma}}_m$ via (18). The statistic $\hat{\boldsymbol{\rho}}$ then depends on the choice of a consistent estimator for $\gamma_0$. In addition, the test statistic relies on an estimator to $w$. We consider two estimators for the couple $(\gamma_0, w)$. First, we can use the estimator $\hat{\gamma}_0$ in (19) and $\hat{w}_1$ in (23). Alternatively, we can use the EVE estimator $\hat{\alpha}$ (24) for $\gamma_0$ and the associated $\hat{w}_2$ for $w$. These two options lead to two test statistics, each of which follows a $\chi_m^2$ distribution asymptotically under the null hypothesis, as guaranteed by Theorem 3. We refer to these two test statistics as SIP 1 and SIP 2, respectively. The primary difference between SIP1 and SIP2 lies in the estimation of the variance component $\gamma_0$ and the associated parameter $w$. SIP1 is based on the estimator $\hat{\gamma}_0$ in (19), which remains consistent under $m$-dependence assumptions, and is therefore more robust to model misspecification. In contrast, SIP2 employs the estimators $(\hat{\alpha}, \hat{\beta})$ in (24), which are derived from a regression formulation and are tailored to the null hypothesis; these estimators typically exhibit lower variance and hence provide improved numerical stability when $m$ is moderately large. As a result, SIP1 and SIP2 reflect a trade-off between robustness and stability. In terms of power, neither method uniformly dominates the other, and their relative performance depends on the underlying dependence structure. Our numerical studies suggest that both procedures perform similarly in most settings, with SIP2 often showing slightly better finite-sample behavior for larger values of $m$.

The choice of the lag parameter $m$ plays an important role in this trade-off. The proposed SIP tests, like classical Box-type portmanteau tests, require specifying $m$, and there is no universally optimal or fully data-driven rule for selecting $m$, even in the stationary setting without mean shifts. In practice, it is customary to choose a small to moderate value (e.g., $m = 4$ or 8), reflecting a balance between capturing serial dependence and maintaining stability of the test statistic. In our framework, the theoretical condition $m + 2 \leq L$ (or $m + 2 \leq L/2$) ensures the validity of the asymptotic results, but it is primarily a sufficient condition and need not be strictly enforced in practice. When $L$ is unknown, we recommend using moderate values of $m$ and, if desired, examining results across several choices of $m$, as is commonly done in classical portmanteau testing. Our numerical studies indicate that the proposed SIP tests are reasonably robust across a range of $m$, while overly large values may lead to increased variability, particularly when mean shifts are frequent.

### 3.6 Shift-Immune ACF plot

The Autocorrelation Function (ACF) plot is a valuable tool that visualizes correlations between observations at successive time lags. In R, the ACF plot is typically generated using the `acf()` function, which displays the strength of autocorrelation at various lags together with 95% significance bounds (under the IID assumption). By examining the ACF plot, practitioners can evaluate stationarity (based on how quickly autocorrelations decay), identify potential seasonal patterns through recurring spikes, and determine significant lags for stationary series. It is worth noting that the ACF plot provides straightforward, reliable information on autocorrelations only if the time series is stationary; otherwise, it serves primarily as a diagnostic tool.

Our framework provides a useful tool to visualize autocorrelation for nonstationary time

series with frequent mean shifts. The main ingredients of an ACF plot include an estimated autocorrelation and a significance bound for each lag of interest. Due to the non-constant nature of $\boldsymbol{\theta}$, it is impossible to estimate the autocovariances consistently without resorting to assuming a further dependence structure (like the $m$-dependence in Tecuapetla-Gómez and Munk (2017)) or to assuming $L$ goes to infinity to allow the asymptotic analysis. As emphasized in the introduction, we intend the exposition in this paper to serve as an initial step of the analysis, and be valid under as mild conditions as possible. Therefore, we do not try to estimate the ACF to generate the plot. Instead, for a pre-specified maximal lag $s$, we use (18) and (19) with $m = h + 2$ to calculate $\hat{\gamma}_0$, $\hat{\gamma}_h$, and $\tilde{\rho}_h = \hat{\gamma}_h / \hat{\gamma}_0$ for each $h = 1, \ldots, s$. The significance bounds are derived from the diagonal elements in $\boldsymbol{\Sigma}_{\rho,\hat{w}}$ in (25), with $\hat{w} = \hat{w}_1$, as defined in (23) with $m = s$. The advantage of this approach is that the standard error of $\hat{\rho}_h$ under IID assumption (or more general, $h$-dependent assumption on $\{\varepsilon_i\}$) is $\sqrt{(6 + 4\hat{w})/n}$, independent of $h$, as verified by the following proposition.

**Proposition 8** *Assume the conditions of Theorem 3. The $(m, m)$ entry of the $m \times m$ matrix* $\boldsymbol{\Sigma}_{\rho,\hat{w}}$ *defined in* (25) *is* $6 + 4\hat{w}$.

We then plot $\tilde{\rho}_h$ against $h$ for $h = 1, \ldots, s$ with the corresponding 95% significance bounds $\pm 1.96\sqrt{(6 + 4\hat{w})/n}$. Note that the $\hat{\gamma}_h$ generated this way is actually estimating $\gamma_h - 2\gamma_{h+1} + \gamma_{h+2}$ (see Proposition 6), which is unbiased only when $\gamma_{h+1} = \gamma_{h+2} = 0$. Therefore, the resulting plot is not an ACF plot in the strict sense. Nevertheless, it provides valuable information about autocorrelation patterns in the presence of mean shifts, and so we adopt the term "ACF plot" by a slight abuse of terminology. When the plot reveals more than one significant autocorrelation, the estimated autocorrelation at the highest lag likely gives a faithful estimate.

## 4 Numerical Studies

### 4.1 Simulated data examples

#### 4.1.1 Type I error control

We illustrate the performance of our proposed test procedures using simulated data. We consider three test procedures, including our proposed methods SIP 1, SIP 2, and the classical Box-Pierce test (labeled as 'Box'). We also add two oracle procedures for comparison. Specifically, assuming the true mean is known, the 'oracle' directly applies the Box-Pierce test to the noise sequence. In the pseudo-oracle procedure (labeled as 'p-oracle'), assuming the locations of the change points are known, the Box-Pierce test is applied to the residuals, which are the differences between observed values and the segment-wise sample means. For all test procedures, the order $m$ (the `lag` parameter of the `Box.test()` function in R) is chosen from the set $\{1, 2, 4, 8\}$.

We consider three scenarios for the noise distribution: a standard Gaussian distribution $\varepsilon_i \sim N(0, 1)$, a scaled student's $t$-distribution with 3 degrees of freedom $\varepsilon_i \sim \sqrt{\frac{1}{3}} t_3$, and a translated exponential distribution $\varepsilon_i \sim Exp(1) - 1$. All distributions have a mean of zero and variance of one. For the mean structure, we fix $n = 10,000$, $J = 100$, and $\boldsymbol{\theta} \in \Theta_{20}$. We randomly generate 100 change-point locations, ensuring that $\mathcal{L}(\boldsymbol{\theta}) \geq 20$. Additionally, the segment mean

parameters are drawn independently from a uniform distribution over the interval $[-5, 5]$. This mean vector is generated and used in all scenarios. In Table 1, we list the average empirical type I error rates over 1,000 independent replicates.

Table 1: Estimated Type I error rates with 1000 replicates under three noise distributions.

|  |  | $m = 1$ | $m = 2$ | $m = 4$ | $m = 8$ |
|---|---|---|---|---|---|
| Gaussian | SIP 1 | 0.047 | 0.034 | 0.050 | 0.080 |
|  | SIP 2 | 0.050 | 0.031 | 0.044 | 0.048 |
|  | Box | 1.000 | 1.000 | 1.000 | 1.000 |
|  | oracle | 0.050 | 0.057 | 0.044 | 0.048 |
|  | p-oracle | 0.172 | 0.233 | 0.322 | 0.412 |
| exponential | SIP 1 | 0.052 | 0.042 | 0.060 | 0.103 |
|  | SIP 2 | 0.044 | 0.040 | 0.049 | 0.048 |
|  | Box | 1.000 | 1.000 | 1.000 | 1.000 |
|  | oracle | 0.045 | 0.058 | 0.052 | 0.054 |
|  | p-oracle | 0.155 | 0.201 | 0.274 | 0.421 |
| student's $t$ | SIP 1 | 0.046 | 0.045 | 0.053 | 0.085 |
|  | SIP 2 | 0.043 | 0.045 | 0.051 | 0.059 |
|  | Box | 1.000 | 1.000 | 1.000 | 1.000 |
|  | oracle | 0.048 | 0.046 | 0.047 | 0.043 |
|  | p-oracle | 0.174 | 0.215 | 0.260 | 0.407 |

We observe that the Box-Pierce test consistently yields a type I error rate of 1, indicating the significant impact of frequent mean shifts on the test for serial correlation. In contrast, the SIP methods demonstrate excellent control of the type I error across different noise distributions, suggesting that their performance is robust to different noise characteristics, including heavy-tailed or asymmetric distributions. This behavior is consistent with the Box-Pierce test for stationary time series. The SIP 1 test controls the type I error well for most considered values of $m$, though some inflation appears when $m = 8$. This is primarily due to the instability of the variance estimator $\hat{\gamma}_0$ for larger values of $m$, which compromises the type I error control. On the other hand, the SIP 2 test performs as effectively as the oracle procedure across all $m$ values.

A noteworthy finding is that even when the true locations of the change points are known, the pseudo-oracle procedure fails to control the type I error. This failure arises because the demeaning step introduces significant autocorrelation in the presence of frequent mean shifts. This suggests that, even when change points can be accurately detected, the SIP method remains the preferred choice for testing serial correlation.

### 4.1.2 Power Analysis

We illustrate the statistical power of the proposed SIP tests and compare them with the oracle procedure. We use the same mean vector $\boldsymbol{\theta}$ as in the type I error analysis, and consider three noise structures detailed below, with $\{z_i\}$ drawn IID from the standard normal distribution.

Noise structure 1: a first-order moving-average model (MA(1)) where $\varepsilon_i = z_i + \omega z_{i-1}$.

Noise structure 2: a fourth-order moving-average model (MA(4)) where $\varepsilon_i = z_i + \sum_{j=1}^{4} \omega_j z_{i-j}$.

Noise structure 3: a first-order auto-regressive model (AR(1)) where $\varepsilon_i = \phi \varepsilon_{i-1} + z_i$.

For the MA(1) case, we vary $\omega$ in $\{-0.1, -0.05, -0.025, 0.025, 0.05, 0.1\}$ to examine the impact of the sign and magnitude of autocorrelation. For the MA(4) case, we consider three scenarios for $\boldsymbol{\omega} = (\omega_1, \ldots, \omega_4)$: $(0.5, 0.4, 0.3, 0.2)$, $(0.1, 0.1, 0.5, -0.4)$, and $(0, 0.1, 0, -0.8)$, with corresponding autocorrelations $\boldsymbol{\rho}$ of $(0.571, 0.409, 0.260, 0.130)$, $(0.028, 0.077, 0.322, 0.284)$, and $(0, 0.012, 0, -0.485)$. For the AR(1) case, we select $\phi$ from $\{-0.1, -0.05, -0.025, 0.025, 0.05, 0.1\}$. We report the average power of the SIP methods and the oracle procedure using 1,000 replicates in Tables 2-4.

Table 2: Average power under MA(1) noise with 1,000 replicates and varying $\omega$.

|  | method | $m = 1$ | $m = 2$ | $m = 4$ | $m = 8$ |
|---|---|---|---|---|---|
| $\omega = -0.1$ | SIP 1 | 0.929 | 0.988 | 0.999 | 0.998 |
|  | SIP 2 | 0.919 | 0.987 | 0.999 | 0.999 |
|  | oracle | 1.000 | 1.000 | 1.000 | 1.000 |
| $\omega = -0.05$ | SIP 1 | 0.469 | 0.598 | 0.675 | 0.644 |
|  | SIP 2 | 0.433 | 0.578 | 0.652 | 0.655 |
|  | oracle | 0.996 | 0.993 | 0.986 | 0.966 |
| $\omega = -0.025$ | SIP 1 | 0.161 | 0.203 | 0.224 | 0.216 |
|  | SIP 2 | 0.142 | 0.176 | 0.196 | 0.184 |
|  | oracle | 0.720 | 0.599 | 0.474 | 0.352 |
| $\omega = 0.025$ | SIP 1 | 0.145 | 0.161 | 0.165 | 0.161 |
|  | SIP 2 | 0.172 | 0.178 | 0.180 | 0.148 |
|  | oracle | 0.671 | 0.567 | 0.452 | 0.350 |
| $\omega = 0.05$ | SIP 1 | 0.491 | 0.606 | 0.671 | 0.635 |
|  | SIP 2 | 0.531 | 0.632 | 0.706 | 0.687 |
|  | oracle | 0.998 | 0.998 | 0.987 | 0.964 |
| $\omega = 0.1$ | SIP 1 | 0.984 | 0.998 | 1.000 | 0.999 |
|  | SIP 2 | 0.987 | 0.998 | 1.000 | 1.000 |
|  | oracle | 1.000 | 1.000 | 1.000 | 1.000 |

Table 3: Average power under MA(4) noise with 1,000 replicates and three scenarios: Scenario 1, $\boldsymbol{\omega} = (0.5, 0.4, 0.3, 0.2)$; Scenario 2, $\boldsymbol{\omega} = (0.1, 0.1, 0.5, -0.4)$, and Scenario 3, $\boldsymbol{\omega} = (0, 0.1, 0, -0.8)$.

|  | method | $m = 1$ | $m = 2$ | $m = 4$ | $m = 8$ |
|---|---|---|---|---|---|
| Scenario 1 | SIP 1 | 0.249 | 0.874 | 1.000 | 1.000 |
|  | SIP 2 | 0.282 | 0.918 | 1.000 | 1.000 |
|  | oracle | 1.000 | 1.000 | 1.000 | 1.000 |
| Scenario 2 | SIP 1 | 1.000 | 1.000 | 1.000 | 1.000 |
|  | SIP 2 | 1.000 | 1.000 | 1.000 | 1.000 |
|  | oracle | 0.741 | 1.000 | 1.000 | 1.000 |
| Scenario 3 | SIP 1 | 0.104 | 1.000 | 1.000 | 1.000 |
|  | SIP 2 | 0.078 | 1.000 | 1.000 | 1.000 |
|  | oracle | 0.097 | 0.143 | 1.000 | 1.000 |

We observe that the SIP tests reliably uncover nontrivial autocorrelation unless it is extremely weak. Specifically, the SIP tests achieve power close to 1 for $\omega = \pm 0.1$ in the MA(1) case and $\phi = \pm 0.1$ in the AR(1) case. For MA(4) noise, we showcase three distinct patterns. In scenario 1, the SIP tests perform poorly with insufficiently large $m$. In scenario 2, the SIP tests outperform the oracle procedure, which suffers at $m = 1$ due to weak lag-1 correlation. In

Table 4: Average power under AR(1) noise with 1,000 replicates and varying $\phi$.

|  | method | $m = 1$ | $m = 2$ | $m = 4$ | $m = 8$ |
|---|---|---|---|---|---|
| $\phi = -0.1$ | SIP 1 | 0.991 | 0.999 | 1.000 | 1.000 |
|  | SIP 2 | 0.989 | 0.999 | 1.000 | 1.000 |
|  | oracle | 1.000 | 1.000 | 1.000 | 1.000 |
| $\phi = -0.05$ | SIP 1 | 0.557 | 0.678 | 0.728 | 0.653 |
|  | SIP 2 | 0.529 | 0.651 | 0.716 | 0.670 |
|  | oracle | 0.999 | 0.997 | 0.986 | 0.974 |
| $\phi = -0.025$ | SIP 1 | 0.204 | 0.226 | 0.216 | 0.222 |
|  | SIP 2 | 0.174 | 0.216 | 0.201 | 0.173 |
|  | oracle | 0.713 | 0.628 | 0.484 | 0.375 |
| $\phi = 0.025$ | SIP 1 | 0.121 | 0.166 | 0.185 | 0.180 |
|  | SIP 2 | 0.143 | 0.184 | 0.199 | 0.160 |
|  | oracle | 0.706 | 0.593 | 0.489 | 0.372 |
| $\phi = 0.05$ | SIP 1 | 0.415 | 0.557 | 0.626 | 0.571 |
|  | SIP 2 | 0.458 | 0.595 | 0.659 | 0.600 |
|  | oracle | 1.000 | 0.998 | 0.988 | 0.967 |
| $\phi = 0.1$ | SIP 1 | 0.923 | 0.984 | 0.999 | 0.998 |
|  | SIP 2 | 0.935 | 0.990 | 1.000 | 0.998 |
|  | oracle | 1.000 | 1.000 | 1.000 | 1.000 |

scenario 3, all methods fail at $m = 1$, but the SIP tests outperform the oracle when $m = 2$. The oracle procedure lacks power when autocorrelations are weak and $m$ is too small because it discards covariance information beyond lag $m$. In contrast, in SIP tests, a smaller $m$ does not necessarily lead to power loss as shown in scenarios 2 and 3, since higher-lag covariance is implicitly incorporated into the test procedure.

In summary, we recommend $m = 4$ as the default choice for both SIP tests. We slightly favor the SIP 2 test for larger $m$ due to its more accurate control of type I error.

## 4.2 Real Data Analysis

We apply the SIP tests to synthesized RNA nanopore sequencing reads from Wang et al. (2024). The dataset comprises 900 sequences of integers representing ionic current levels, with an average sequence length exceeding 30,000. Figure 1 in Section 1 illustrates two examples of nanopore sequencing data, showcasing frequent mean shifts. While some of these shifts are visually apparent, accurately identifying the exact locations of all change points remains challenging.

We apply both SIP tests with $m \in \{1, 2, 4, 8\}$ to the 900 sequences. Table 5 summarizes the test results, which indicate significant serial correlation in all sequences at the 5% level. There are only 32 insignificant $p$-values, which occur when $m = 1$. Overall, this provides strong evidence of significant autocorrelation in nanopore sequencing data. To gain further insight into the autocorrelation structure, we visualize the low-order autocorrelations using our proposed SIP-ACF plot on the data shown in Figure 1, where the two sequences are from the data in Wang et al. (2024) labeled as 33 and 39, with UUIDs `read_0836bca7-a0b7-412d-9e9a-8abe0ee813df` and `read_0abeb32d-c351-427b-ae41-ddf46ba1302d`. The SIP-ACF plots, presented in the right column of Figure 2, reveal small but significant positive lag-1 autocorrelations. Similar patterns

are observed in most of the nanopore sequences examined. We also detect small lag-2 to lag-4 autocorrelations, with significance levels varying among sequences. In contrast, the classical ACF plots, shown in the left column of Figure 2, completely fail to capture the autocorrelation structure due to the frequent mean shifts.

Table 5: Summary of test results on nanopore sequencing data across lags.

|       | Method | Reject $H_0$ | Accept $H_0$ | Average p-value | Maximum p-value |
|-------|--------|--------------|--------------|-----------------|-----------------|
| $m=1$ | SIP 1  | 868          | 32           | 0.010           | 0.930           |
|       | SIP 2  | 868          | 32           | 0.011           | 0.930           |
| $m=2$ | SIP 1  | 900          | 0            | $9.415 \times 10^{-5}$ | 0.030    |
|       | SIP 2  | 900          | 0            | $7.774 \times 10^{-5}$ | 0.033    |
| $m=4$ | SIP 1  | 900          | 0            | $1.408 \times 10^{-5}$ | 0.012    |
|       | SIP 2  | 900          | 0            | $1.032 \times 10^{-5}$ | $8.841 \times 10^{-3}$ |
| $m=8$ | SIP 1  | 900          | 0            | $9.811 \times 10^{-9}$ | $4.752 \times 10^{-6}$ |
|       | SIP 2  | 900          | 0            | $7.247 \times 10^{-9}$ | $6.172 \times 10^{-6}$ |

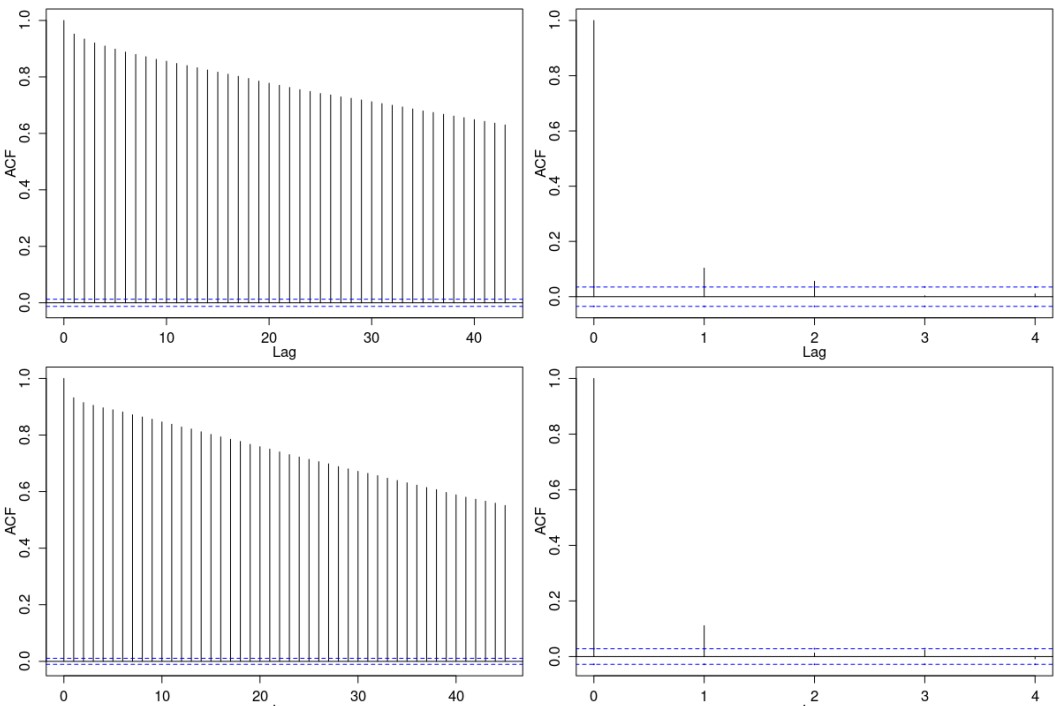

Figure 2: Left: the standard ACF plots for nanopore sequences with id= 33 and 39; right: shift-immune ACF plots for nanopore sequences with id=33 and 39.

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

# A  Technical Proofs

The theoretical results in Sections 2 and 3 are derived in Appendices A.1 and A.2, respectively.

## A.1  Proofs of theoretical results in Sections 2

**Proof of Proposition 1:** Simple calculations show

$$
\begin{aligned}
\mathrm{E}\left(\boldsymbol{X}^{\top}\boldsymbol{A}\boldsymbol{X}\right) &= \mathrm{E}\left[(\boldsymbol{\theta}+\boldsymbol{\varepsilon})^{\top}\boldsymbol{A}(\boldsymbol{\theta}+\boldsymbol{\varepsilon})\right] \\
&= \mathrm{E}\left(\boldsymbol{\theta}^{\top}\boldsymbol{A}\boldsymbol{\theta} + \boldsymbol{\theta}^{\top}\boldsymbol{A}\boldsymbol{\varepsilon} + \boldsymbol{\varepsilon}^{\top}\boldsymbol{A}\boldsymbol{\theta} + \boldsymbol{\varepsilon}^{\top}\boldsymbol{A}\boldsymbol{\varepsilon}\right) \\
&= \boldsymbol{\theta}^{\top}\boldsymbol{A}\boldsymbol{\theta} + \mathrm{E}\left(\boldsymbol{\varepsilon}^{\top}\boldsymbol{A}\boldsymbol{\varepsilon}\right) \\
&= \sum_{h=1-n}^{n-1}(n-|h|)a_{|h|}\gamma_h + \boldsymbol{\theta}^{\top}\boldsymbol{A}\boldsymbol{\theta}.
\end{aligned}
$$

$\square$

**Proof of Theorem 1**: In the trivial case $L = 1$, $\Theta_1 = \mathbb{R}^n$. The quadratic form $\boldsymbol{\theta}^{\top}\boldsymbol{A}\boldsymbol{\theta} = 0$ for all $\boldsymbol{\theta}$ if and only if $\boldsymbol{A} = \boldsymbol{0}$, i.e. $a_i = 0$ for all $0 \le i < n$. It is easy to check that the equations (7)-(10) hold if and only if $a_i = 0$ for all $0 \le i < n$. Next we work on the more interesting cases.

We prove the necessity first. We are going to plug various $\boldsymbol{\theta} \in \Theta_L$ in the equality $\boldsymbol{\theta}^{\top}\boldsymbol{A}\boldsymbol{\theta} = 0$ to achieve the equations (7)-(10). Let $\mathbf{1}_t$ and $\mathbf{0}_t$ be the $t$-dimensional vectors $(1,\ldots,1)^{\top}$ and $(0,\ldots,0)^{\top}$ respectively. Let $\boldsymbol{\theta} = (\mathbf{1}_L^{\top}, \mathbf{0}_{n-L}^{\top})^{\top}$. $\boldsymbol{\theta}^{\top}\boldsymbol{A}\boldsymbol{\theta}$ calculates the sum of all entries of the top-left $L \times L$ submatrix of $\boldsymbol{A}$. So the equality $\boldsymbol{\theta}^{\top}\boldsymbol{A}\boldsymbol{\theta} = 0$ implies

$$
La_0 + 2(L-1)a_1 + \cdots + 4a_{L-2} + 2a_{L-1} = 0.
$$

With $\boldsymbol{\theta} = (\mathbf{1}_{L+1}^{\top}, \mathbf{0}_{n-L-1}^{\top})^{\top}$, the equality $\boldsymbol{\theta}^{\top}\boldsymbol{A}\boldsymbol{\theta} = 0$ implies

$$
(L+1)a_0 + 2La_1 + \cdots + 4a_{L-1} + 2a_L = 0. \tag{27}
$$

It yields (7) by taking the difference of (27) and its preceding equation. Note that both $(\mathbf{1}_L^{\top}, \mathbf{0}_{n-L}^{\top})^{\top}$ and $(\mathbf{1}_{L+1}^{\top}, \mathbf{0}_{n-L-1}^{\top})^{\top}$ are in $\Theta_L$ by the condition $L < n/2$. Now if we multiply

(7) by $(L+1)$ and then subtract (27) from it, we get (8). The equation (9) is not necessary when $n = 2L + 1$. When $n \geq 2L + 2$, $(\mathbf{1}_{L+2}^\top, \mathbf{0}_{n-L-2}^\top)^\top \in \Theta_L$. We can continue the process and get

$$(L+2)a_0 + 2(L+1)a_1 + \cdots + 4a_L + 2a_{L+1} = 0. \tag{28}$$

Subtracting the sum of (27) and (7) from (28), we obtain $a_{L+1} = 0$. Following the same procedure with $\boldsymbol{\theta} = (\mathbf{1}_{L+3}^\top, \mathbf{0}_{n-L-3}^\top)^\top, \ldots, (\mathbf{1}_{n-L}^\top, \mathbf{0}_L^\top)^\top$, we can derive $a_{L+1} = \cdots = a_{n-L-1} = 0$. Therefore, (9) holds.

Finally, we employ $\boldsymbol{\theta} = \mathbf{1}_n$ in the equality $\boldsymbol{\theta}^\top \boldsymbol{A} \boldsymbol{\theta} = 0$ and obtain

$$na_0 + 2(n-1)a_1 + \cdots + 4a_{n-2} + 2a_{n-1} = 0$$

which yields (10) by applying the results (7)-(9).

Now we show the sufficiency. For any $\boldsymbol{\theta} \in \Theta_L$, as in (2), we can write $\boldsymbol{\theta} = \sum_{j=1}^{J+1} \mu_j \boldsymbol{\theta}_j$ where $\boldsymbol{\theta}_1 = (\mathbf{1}_{\tau_1}^\top, \mathbf{0}_{n-\tau_1}^\top)^\top$, $\boldsymbol{\theta}_j = (\mathbf{0}_{\tau_j}^\top, \mathbf{1}_{\tau_{j+1}-\tau_j}^\top, \mathbf{0}_{n-\tau_{j+1}}^\top)^\top$, for $j = 2, \ldots, J$, and $\boldsymbol{\theta}_{J+1} = (\mathbf{0}_{\tau_J}^\top, \mathbf{1}_{n-\tau_J}^\top)^\top$.

$$\boldsymbol{\theta}^\top \boldsymbol{A} \boldsymbol{\theta} = \sum_{j=1}^{J+1} \sum_{\ell=1}^{J+1} \mu_j \mu_\ell \boldsymbol{\theta}_j^\top \boldsymbol{A} \boldsymbol{\theta}_\ell.$$

It suffices to show $\boldsymbol{\theta}_j^\top \boldsymbol{A} \boldsymbol{\theta}_\ell = 0$ for all $j$, $\ell$. It is straightforward to see $(\sum_{\ell=1}^j \boldsymbol{\theta}_\ell)^\top \boldsymbol{A} (\sum_{\ell=1}^j \boldsymbol{\theta}_\ell) = 0$ for $1 \leq j \leq J+1$ from the proof of the necessity part. In particular, we have $\boldsymbol{\theta}_1^\top \boldsymbol{A} \boldsymbol{\theta}_1 = 0$ and $(\boldsymbol{\theta}_1 + \boldsymbol{\theta}_2)^\top \boldsymbol{A} (\boldsymbol{\theta}_1 + \boldsymbol{\theta}_2) = 0$, where the latter gives $\boldsymbol{\theta}_1^\top \boldsymbol{A} \boldsymbol{\theta}_1 + 2\boldsymbol{\theta}_1^\top \boldsymbol{A} \boldsymbol{\theta}_2 + \boldsymbol{\theta}_2^\top \boldsymbol{A} \boldsymbol{\theta}_2 = 0$. Note that $\boldsymbol{\theta}_2^\top \boldsymbol{A} \boldsymbol{\theta}_2 = 0$ as $\boldsymbol{A}$ is Toeplitz. (The fact that $\boldsymbol{A}$ is Toeplitz implies that $\boldsymbol{\theta}_2^\top \boldsymbol{A} \boldsymbol{\theta}_2$ won't change if we move all the 1s in $\boldsymbol{\theta}_2$ to the top coordinates.) Therefore, we can claim $\boldsymbol{\theta}_1^\top \boldsymbol{A} \boldsymbol{\theta}_2 = 0$. By the same argument, we can get $\boldsymbol{\theta}_j^\top \boldsymbol{A} \boldsymbol{\theta}_{j+1} = 0$. Moreover, it follows (9) that $\boldsymbol{\theta}_j^\top \boldsymbol{A} \boldsymbol{\theta}_\ell = 0$ for $\ell - j > 1$. $\qquad\square$

**Proof of Proposition 2**: The quadratic form $\boldsymbol{X}^\top \boldsymbol{A} \boldsymbol{X}$ described in Theorem 1 is invariant under a global mean shift if and only if $\boldsymbol{A}\mathbf{1} = \mathbf{0}$, i.e., the row sums of $\boldsymbol{A}$ are all zero. The sufficiency in Proposition 2 is then obvious due to equation (7). To show the necessity, we first observe that the sum of $(L+1)$-th row of $\boldsymbol{A}$, given equation (9), is

$$a_0 + 2a_1 + \cdots + 2a_{L-1} + a_L + a_{n-L} = 0,$$

which implies $a_{n-L} = a_L$ in view of (7). Next we move up to the $L$-th row:

$$a_0 + 2a_1 + \cdots + 2a_{L-2} + a_{L-1} + a_L + a_{n-L} + a_{n-L+1} = 0,$$

which, together with (7) and $a_{n-L} = a_L$, implies that $a_{n-L+1} = a_{L-1}$. The proof of necessity is completed by moving upward of the equation $\boldsymbol{A}\mathbf{1} = \mathbf{0}$ row by row till the top. $\qquad\square$

**Proof of Proposition 3**: We can express $T_h$ as

$$T_h = \sum_{i=1}^n (X_i - X_{i+h})^2 = 2\sum_{i=1}^n X_i^2 - 2\sum_{i=1}^n X_i X_{i+h}, \quad h = 1, \ldots, L. \tag{29}$$

That is, each $T_h$ can be rendered as a quadratic form using the circulant matrix induced by

$(2, 0, \ldots, 0, -1, 0, \ldots, 0, -1, 0, \ldots, 0)$, where the two $-1$ appear as the $(h+1)$-th and $(n-h+1)$-th entries. It is straightforward to verify (13) with equation (7). $\qquad\square$

**Proof of Proposition 4**: Since $\boldsymbol{A} \in \mathcal{A}_L$, it holds that $\boldsymbol{\theta}^\top \boldsymbol{A} \boldsymbol{\theta} = 0$. The proposition is then an immediate consequence of Proposition 1. $\qquad\square$

## A.2   Proofs of theoretical results in Sections 3

**Proof of Proposition 5**: Since $\{\varepsilon_i\}$ is $m$-dependent, by Proposition 1 and Theorem 1

$$\mathrm{E}\hat{\gamma}_h = 2(n-h)a_{hh}\gamma_h = (1 - h/n)\gamma_h.$$

According to (13), the $\hat{\gamma}_0$ defined in (19) can be represented as $\boldsymbol{X}^\top \boldsymbol{A}_0 \boldsymbol{X}$, where $\boldsymbol{A}_0$ takes the form (5) with the first row

$$\left(1/n, 0, \ldots, 0, -\frac{m+2}{2n}, \frac{m+1}{2n}, 0, \ldots, 0, \frac{m+1}{2n}, -\frac{m+2}{2n}, 0, \ldots, 0\right),$$

where the five nonzero values are at the 1st, $(m+2)$-th, $(m+3)$-th, $(n-(m+2))$-th, and $(n-(m+1))$-th entries. The expectation of $\hat{\gamma}_0$ is therefore $\gamma_0$ again by Proposition 1. $\qquad\square$

We shall defer the proof of Proposition 6 after that of Theorem 2, as it could make use of some arguments in the latter. The proof of Theorem 2 relies on the following lemma which establishes the asymptotic property of $\{T_h\}$ as defined in (29).

**Lemma 1** *Under conditions 1, 2, and 3, and the null hypothesis that $\varepsilon_1, \ldots, \varepsilon_n$ are IID, if $K \leq L/2$, then $\boldsymbol{T}_K = (T_1, ..., T_K)^\top$ is asymptotically normal as $n \to \infty$,*

$$\boldsymbol{\Sigma}_{K,w}^{-1/2}\sqrt{n}(\boldsymbol{T}_K/n - \boldsymbol{\nu}_K) \xrightarrow{D} \mathcal{N}(\boldsymbol{0}_K, \boldsymbol{I}_K), \tag{30}$$

*where*

$$\boldsymbol{\Sigma}_{K,w} = 4\gamma_0^2 \left[\boldsymbol{I}_K + (\kappa_4 - 1)\boldsymbol{1}_K\boldsymbol{1}_K^\top + 2w\boldsymbol{H}_K\right], \quad \boldsymbol{\nu}_K = \gamma_0(2 \cdot \boldsymbol{1}_K + w\boldsymbol{\eta}_K).$$

*Moreover, the range of $K$ can be relaxed to $K \leq L$, in which case*

$$\lim_{n\to\infty} \boldsymbol{\Sigma}_{K,2w}^{-1/2}\sqrt{n}(\boldsymbol{T}_K/n - \boldsymbol{\nu}_K) \preceq \mathcal{N}(\boldsymbol{0}_K, \boldsymbol{I}_K). \tag{31}$$

**Proof of Lemma 1.** By Proposition 2.1 in Hao et al. (2023), the expectation and covariance of $n^{-1}\boldsymbol{T}_K$ are $\boldsymbol{\nu}_K$ and $n^{-1}\boldsymbol{\Sigma}_{K,w}$, respectively, under the null hypothesis. In particular, the variance of $T_k$ is $4n\gamma_0^2(\kappa_4 + 2kw)$. For ease of presentation, we will prove a central limit theorem for $(2\gamma_0)^{-1} \cdot [n(\kappa_4 + 2kw)]^{-1/2} \cdot (T_k - n\nu_k)$ for every $1 \leq k \leq K$. The central limit theorem for $\boldsymbol{T}_K$ can be established along similar lines with an application of the Cramér-Wold device.

Write $T_k$ as

$$
\begin{aligned}
T_k &= \sum_{i=1}^{n}(X_i - X_{i+k})^2 \\
&= \sum_{i=1}^{n}(\theta_i - \theta_{i+k} + \varepsilon_i - \varepsilon_{i+k})^2 \\
&= \sum_{i=1}^{n}(\theta_i - \theta_{i+k})^2 + 2\sum_{i=1}^{n}(\theta_i - \theta_{i+k})(\varepsilon_i - \varepsilon_{i+k}) + \sum_{i=1}^{n}(\varepsilon_i - \varepsilon_{i+k})^2 \\
&= n\gamma_0(2 + kw) + 2\sum_{i=1}^{n}(\theta_i - \theta_{i+k})(\varepsilon_i - \varepsilon_{i+k}) + \sum_{i=1}^{n}\left[(\varepsilon_i - \varepsilon_{i+k})^2 - 2\gamma_0\right] \\
&= n\gamma_0(2 + kw) + 2\sum_{i=1}^{n}d_{ki}\varepsilon_i + 2\sum_{i=1}^{n}(\varepsilon_i^2 - \gamma_0) - 2\sum_{i=1}^{n}\varepsilon_i\varepsilon_{i+k},
\end{aligned}
$$

where all the indices are modulo $n$, and $d_{ki} = \mu_j - \mu_{j+1}$ for $\tau_j - k < i \le \tau_j$, $d_{ki} = \mu_{j+1} - \mu_j$ for $\tau_j < i \le \tau_j + k$, and $d_{ki} = 0$ otherwise. We shall apply the martingale central limit theorem (see Corollary 3.1 of Hall and Heyde (1980)). Note that since the number, location and magnitude of the change-points are allowed to change with the sample size, we are using the triangular-array version of the martingale and its central limit theorem. In order to have a more homogeneous form of the martingale differences, we will use indices that are no longer modulo $n$ for the rest of this proof, and modify $T_k$ slightly by considering

$$
\tilde{T}_k = n\gamma_0(2 + kw) + 2\sum_{i=1}^{n}d_{ki}\varepsilon_i + 2\sum_{i=1}^{n}(\varepsilon_i^2 - \gamma_0) - 2\sum_{i=1}^{n}\varepsilon_i\varepsilon_{i-k}, \tag{32}
$$

where $\varepsilon_0, \varepsilon_{-1}, \ldots, \varepsilon_{1-k}$ are IID with other $\varepsilon_i$'s. It is clear that $\tilde{T}_k - T_k = O_P(1)$, so it suffices to have a central limit theorem for $\tilde{T}_k$. Let

$$
D_{ni} := (2\gamma_0)^{-1} \cdot [n(\kappa_4 + 2kw)]^{-1/2}[2d_{ki}\varepsilon_i + 2(\varepsilon_i^2 - \gamma_0) - \varepsilon_i\varepsilon_{i-k}],
$$

and $\mathcal{F}_i$ be the $\sigma$-field generated by $\{\varepsilon_{1-k}, \ldots, \varepsilon_0, \varepsilon_1, \varepsilon_2, \ldots, \varepsilon_i\}$, then $\{D_{ni}\}$ is a martingale difference sequence with respect to the filtration $\{\mathcal{F}_i\}$. To apply the martingale central limit theorem, it suffices to show that

$$
\sum_{i=1}^{n}\mathrm{E}\left\{D_{ni}^2 I[|D_{ni}| \ge \epsilon]\right\} \to 0, \quad \forall \epsilon > 0.
$$

Let $u_i := 4\varepsilon_i^2 + 4(\varepsilon_i^2 - \gamma_0)^2 + \varepsilon_i^2\varepsilon_{i-k}^2$, then the preceding sum is upper bounded by

$$
\begin{aligned}
&\left[4n\gamma_0^2(\kappa_4 + 2kw)\right]^{-1}\sum_{i=1}^{n}\mathrm{E}\left\{(d_{ki}^2 + 2)u_i I\left[|u_i| \ge \frac{\epsilon^2 \cdot 4n\gamma_0^2(\kappa_4 + 2kw)}{(d_{ki}^2 + 2)}\right]\right\} \\
&\le \frac{2n(1 + \gamma_0 wk)}{4n\gamma_0^2(\kappa_4 + 2kw)}\mathrm{E}\left\{u_i I\left[|u_i| \ge \frac{\epsilon^2 \cdot 4n\gamma_0^2(\kappa_4 + 2kw)}{(\max_i d_{ki}^2 + 2)}\right]\right\}.
\end{aligned}
$$

Condition 3 entails that $\max_i d_{ki}^2 = o(n)$, so the expectation in the preceding equation goes to zero as $n \to \infty$, and the proof of (30) is complete.

The second statement (31) for the case $K \le L$ follows the fact that the covariance matrix of $\boldsymbol{T}_K/\sqrt{n}$ is upper bounded by $\boldsymbol{\Sigma}_{K,2w}$, see the proof of the second part of Theorem 2.4 in Hao et al. (2023). $\square$

**Proof of Theorem 2**: As $\hat{\gamma}_0$ is the linear combination of $T_{m+1}$ and $T_{m+2}$, whose asymptotic joint distribution is shown in Lemma 1, it is straightforward to show the asymptotic distribution of $\hat{\gamma}_0$ as

$$\frac{\sqrt{n}(\hat{\gamma}_0 - \gamma_0)}{\gamma_0 \sqrt{[\kappa_4 + 2(m+1)(m+2)(1+w)]}} \to \mathcal{N}(0,1),$$

Note that Condition 3 implies $w = W(\boldsymbol{\theta})/(n\gamma_0) \le n \cdot \max_j\{(\mu_j - \mu_{j+1})^2\}/(n\gamma_0) = o(n)$. Together with the finite fourth moment condition, this implies that the variance of $\hat{\gamma}_0$ approaches zero as $n \to \infty$. Therefore we can conclude that $\hat{\gamma}_0$ converges in probability to $\gamma_0$ by Chebyshev's inequality.

Similarly, the asymptotic normality of $\hat{\boldsymbol{\gamma}}$ is implied by (18), (19) and Lemma 1, as $\hat{\boldsymbol{\gamma}}$ is a linear function of $\boldsymbol{T}_{m+2}$. In particular, $\hat{\boldsymbol{\gamma}} = \boldsymbol{R}\boldsymbol{T}_{m+2}/(2n)$, where

$$\boldsymbol{R} = \begin{pmatrix} -1 & 0 & \cdots & 0 & m+1 & -m \\ 0 & -1 & \cdots & 0 & m & -(m-1) \\ \vdots & \vdots & \ddots & \vdots & \vdots & \vdots \\ 0 & 0 & \cdots & -1 & 2 & -1 \end{pmatrix} = (-\boldsymbol{I}_m, (m+2)\boldsymbol{1}_m - \boldsymbol{\eta}_m, \boldsymbol{\eta}_m - (m+1)\boldsymbol{1}_m).$$

By Lemma 1, the covariance matrix $\boldsymbol{\Sigma}_{m+2,w}$ of $\boldsymbol{T}_{m+2}/n$ could be decomposed as

$$\boldsymbol{\Sigma}_{m+2,w} = 4n^{-1}\gamma_0^2 \begin{pmatrix} \boldsymbol{\Sigma}_m/(4n^{-1}\gamma_0^2) & (\kappa_4 - 1)\boldsymbol{1}_m + 2w\boldsymbol{\eta}_m & (\kappa_4 - 1)\boldsymbol{1}_m + 2w\boldsymbol{\eta}_m \\ (\kappa_4 - 1)\boldsymbol{1}_m^\top + 2w\boldsymbol{\eta}_m^\top & \kappa_4 + 2(m+1)w & (\kappa_4 - 1) + 2(m+1)w \\ (\kappa_4 - 1)\boldsymbol{1}_m^\top + 2w\boldsymbol{\eta}_m^\top & (\kappa_4 - 1) + 2(m+1)w & \kappa_4 + 2(m+2)w \end{pmatrix}.$$

Therefore, $\boldsymbol{\Sigma}_\gamma = \boldsymbol{R}\boldsymbol{\Sigma}_{m+2,w}\boldsymbol{R}^\top/4$, which is further detailed as

$$\begin{aligned} \boldsymbol{\Sigma}_\gamma =& \boldsymbol{R}\boldsymbol{\Sigma}_{m+2,w}\boldsymbol{R}^\top/4 \\ =& \gamma_0^2 \{\boldsymbol{I}_m + \left[(2m^2 + 6m + 5) + 2(m^2 + 3m + 2)w\right]\boldsymbol{1}_m\boldsymbol{1}_m^\top \\ &- [(2m+3) + 2(m+2)w](\boldsymbol{\eta}_m\boldsymbol{1}_m^\top + \boldsymbol{1}_m\boldsymbol{\eta}_m^\top) \\ &+ (2+2w)\boldsymbol{\eta}_m\boldsymbol{\eta}_m^\top + 2w\boldsymbol{H}_m\}. \end{aligned}$$

The proof is complete. $\square$

**Remark 1.** We would like to point out that while the kurtosis $\kappa_4$ of $\varepsilon_i$ appears in the covariance matrix $\boldsymbol{\Sigma}_{m+2,w}$ of $\boldsymbol{T}_{m+2}$, it no longer shows up in the covariance matrix $\boldsymbol{\Sigma}_\gamma$ of $\hat{\boldsymbol{\gamma}}$. This is because all the square terms $\varepsilon_i^2$ are eliminated from $\hat{\boldsymbol{\gamma}}$ in order to remove $\gamma_0$ from its expectation. As a result, the fourth moment does not appear in the variance calculation.

**Proof of Proposition 6**: We divide the proof into two steps: first

$$\lim_{n\to\infty} \mathcal{P}_{m+2}[\mathrm{E}\boldsymbol{T}_{m+2}] = \mathcal{P}_{m+2}\boldsymbol{\gamma}_{m+2}, \tag{33}$$

and next,
$$\mathcal{P}_{m+2}\boldsymbol{T}_{m+2} - \mathcal{P}_{m+2}[\mathrm{E}\boldsymbol{T}_{m+2}] \xrightarrow{P} \boldsymbol{0}. \tag{34}$$

A direct calculation shows that $\mathrm{E}[T_h/(2n)] = \gamma_0 + hw/2 + (1 - h/n)\gamma_h + (h/n)\gamma_{n-h}$ for $1 \le h \le (m+2)$. According to Theorem 1 and Proposition 4, the expectation of $\mathcal{P}_{m+2}\boldsymbol{T}_{m+2}$ does not involve $\boldsymbol{\theta}$ and $\gamma_0$, and hence (33) follows. To show (34), it suffices to show it for $\tilde{\boldsymbol{T}}_{m+2}$ instead of $\boldsymbol{T}_{m+2}$, where $\tilde{\boldsymbol{T}}_{m+2} := (\tilde{T}_1, \tilde{T}_2, \ldots, \tilde{T}_{m+2})^\top$, and $\tilde{T}_k$ is defined in (32). Again by Theorem 1 and Proposition 4, the terms $\gamma_0$, $w$ and $\varepsilon_i^2$ will all disappear in $\mathcal{P}_{m+2}\tilde{\boldsymbol{T}}_{m+2}$, so it in turn suffices to show that
$$n^{-1}\sum_{i=1}^{n} d_{ki}\varepsilon_i - n^{-1}\sum_{i=1}^{n}(\varepsilon_i\varepsilon_{i-k} - \gamma_k) \xrightarrow{P} 0$$

for every $1 \le k \le m+2$. The convergence of the second term $n^{-1}\sum_{i=1}^{n}(\varepsilon_i\varepsilon_{i-k} - \gamma_k)$ to zero in probability is warranted by the ergodic theorem, under Condition 2. The first term converges to zero in probability due to Condition 3. The proof is complete. $\qquad\square$

The following lemma establishes the consistency of $\hat{w}_1$ and $\hat{w}_2$, which plays a key role in the proof of Theorem 3.

**Lemma 2** *(i) Under conditions 1-3, and $\{\varepsilon_i\}$ is $m$-dependent with $m + 2 \le L$, $\hat{w}_1 - w \xrightarrow{P} 0$.*
*(ii) Under one more condition that $\varepsilon_1, \ldots, \varepsilon_n$ are IID, $\hat{w}_2 - w \xrightarrow{P} 0$.*

**Proof of Lemma 2**: Part (i). When $h > m$, $E(X_i X_{i+h}) = E(X_i)E(X_{i+h}) = \theta_i\theta_{i+h}$. It follows that

$$
\begin{aligned}
E&\left(\sum_{i=1}^{n} X_i X_{i+m+2} - \sum_{i=1}^{n} X_i X_{i+m+1}\right) \\
&= \sum_{i=1}^{n}\theta_i\theta_{i+m+2} - \sum_{i=1}^{n}\theta_i\theta_{i+m+1} \\
&= \frac{1}{2}\sum_{i=1}^{n}(\theta_i - \theta_{i+m+2})^2 - \frac{1}{2}\sum_{i=1}^{n}(\theta_i - \theta_{i+m+1})^2 \\
&= \frac{m+2}{2}W(\boldsymbol{\theta}) - \frac{m+1}{2}W(\boldsymbol{\theta}) \\
&= \frac{1}{2}W(\boldsymbol{\theta}) \\
&= n\gamma_0 w/2.
\end{aligned}
$$

Moreover,

$$
\begin{aligned}
&\mathrm{Var}\left(\sum_{i=1}^{n} X_i X_{i+m+2} - \sum_{i=1}^{n} X_i X_{i+m+1}\right) \\
=&\mathrm{Var}\sum_{i=1}^{n}\left(\theta_i\varepsilon_{i+m+2} + \theta_{i+m+2}\varepsilon_i + \varepsilon_i\varepsilon_{i+m+2} - \theta_i\varepsilon_{i+m+1} - \theta_{i+m+1}\varepsilon_i - \varepsilon_i\varepsilon_{i+m+1}\right) \\
=&\mathrm{Var}\left(\sum_{i=1}^{n}(\theta_i - \theta_{i+1})\varepsilon_{i+m+2} + \sum_{i=1}^{n}(\theta_{i+m+2} - \theta_{i+m+1})\varepsilon_i + \sum_{i=1}^{n}\varepsilon_i\varepsilon_{i+m+2} - \sum_{i=1}^{n}\varepsilon_i\varepsilon_{i+m+1}\right) \\
\le&4\mathrm{Var}\sum_{i=1}^{n}(\theta_i - \theta_{i+1})\varepsilon_{i+m+2} + 4\mathrm{Var}\sum_{i=1}^{n}(\theta_{i+m+2} - \theta_{i+m+1})\varepsilon_i + 4\mathrm{Var}\sum_{i=1}^{n}\varepsilon_i\varepsilon_{i+m+2} + 4\mathrm{Var}\sum_{i=1}^{n}\varepsilon_i\varepsilon_{i+m+1} \\
\le&8W(\boldsymbol{\theta})\gamma_0 + 8n(2m+1)\gamma_0^2.
\end{aligned}
$$

In the last inequality, we use two facts. First, $\sum_{i=1}^{n}(\theta_i - \theta_{i+1})\varepsilon_{i+m+2}$ is a linear combination of at most $J + 1$ IID random variables because of the $m$-dependent assumption, $\boldsymbol{\theta} \in \Theta_L$ and $L > m$. As a result, $\operatorname{Var}\sum_{i=1}^{n}(\theta_i - \theta_{i+1})\varepsilon_{i+m+2} \leq \gamma_0 \sum_{i=1}^{n}(\theta_i - \theta_{i+1})^2 = \gamma_0 W(\boldsymbol{\theta})$. Similarly, we can bound $\operatorname{Var}\sum_{i=1}^{n}(\theta_{i+m+2} - \theta_{i+m+1})\varepsilon_i$. Second, to bound

$$\operatorname{Var}\sum_{i=1}^{n}\varepsilon_i\varepsilon_{i+m+1} = \operatorname{Cov}\left(\sum_{i=1}^{n}\varepsilon_i\varepsilon_{i+m+1}, \sum_{j=1}^{n}\varepsilon_j\varepsilon_{j+m+1}\right),$$

we observe that for each $i$, there are at most $2m + 1$ different $j$ $(i - m \leq j \leq i + m)$ such that $\operatorname{Cov}(\varepsilon_i\varepsilon_{i+m+1}, \varepsilon_j\varepsilon_{j+m+1}) \neq 0$. For those pairs, by the Cauchy-Schwarz inequality,

$$\operatorname{Cov}(\varepsilon_i\varepsilon_{i+m+1}, \varepsilon_j\varepsilon_{j+m+1}) \leq \sqrt{\operatorname{Var}(\varepsilon_i\varepsilon_{i+m+1})\operatorname{Var}(\varepsilon_j\varepsilon_{j+m+1})} = \gamma_0^2.$$

Therefore, we have

$$\operatorname{Var}\sum_{i=1}^{n}\varepsilon_i\varepsilon_{i+m+2} \leq n(2m + 1)\gamma_0^2.$$

Given the calculations of expectation and variance above, we conclude

$$n^{-1}\left(\sum_{i=1}^{n}X_iX_{i+m+2} - \sum_{i=1}^{n}X_iX_{i+m+1}\right)$$

is an unbiased estimator of $\gamma_0 w/2$ with variance at most

$$(8W(\boldsymbol{\theta})\gamma_0 + 8n(1 + 2m)\gamma_0^2)/n^2 = 8\gamma_0^2(w + 1 + 2m)/n,$$

which goes to 0 as $n \to \infty$. As a result,

$$2n^{-1}\left(\sum_{i=1}^{n}X_iX_{i+m+2} - \sum_{i=1}^{n}X_iX_{i+m+1}\right) - \gamma_0 w \xrightarrow{P} 0,$$

and

$$2(n\hat{\gamma}_0)^{-1}\left(\sum_{i=1}^{n}X_iX_{i+m+2} - \sum_{i=1}^{n}X_iX_{i+m+1}\right) - w \xrightarrow{P} 0,$$

for any consistent estimator $\hat{\gamma}_0$.

Part (ii). The property of the least squares estimator $(\hat{\alpha}, \hat{\beta})^\top$ is studied in Hao et al. (2023) where Theorem 2.1 shows that it is unbiased with asymptotic variance $n^{-1}O(1 + w)$. Note that Condition 3 implies $w = W(\boldsymbol{\theta})/(n\gamma_0) \leq n \cdot \max_j\{(\mu_j - \mu_{j+1})^2\}/(n\gamma_0) = o(n)$. Therefore, $(\hat{\alpha}, \hat{\beta})^\top$ is unbiased and its variance goes to 0, which implies $(\hat{\alpha}, \hat{\beta})^\top \xrightarrow{P} (\alpha, \beta)^\top$. By definition $\alpha = \gamma_0 > 0$, so $\hat{w}_2$ is consistent. $\qquad\square$

**Proof of Theorem 3**: For any autocorrelation estimator $\hat{\boldsymbol{\rho}} = \hat{\boldsymbol{\gamma}}/\hat{\gamma}_0$ where $\hat{\gamma}_0$ is consistent, it follows from Theorem 2 and Slutsky's theorem that $\boldsymbol{\Sigma}_{\rho,w}^{-\frac{1}{2}}\sqrt{n}\hat{\boldsymbol{\rho}} \xrightarrow{D} \mathcal{N}(\boldsymbol{0}_m, \boldsymbol{I}_m)$, where $\boldsymbol{\Sigma}_{\rho,w} = \boldsymbol{\Sigma}_{\gamma,w}/\gamma_0^2$.

Now we show $\boldsymbol{\Sigma}_{\rho,\hat{w}}^{-\frac{1}{2}}\boldsymbol{\Sigma}_{\rho,w}^{\frac{1}{2}} \xrightarrow{P} \boldsymbol{I}_m$. We can write $\boldsymbol{\Sigma}_{\rho,\hat{w}}^{-\frac{1}{2}}\boldsymbol{\Sigma}_{\rho,w}^{\frac{1}{2}} = (\boldsymbol{M}_1 + \hat{w}\boldsymbol{M}_2)^{-\frac{1}{2}}(\boldsymbol{M}_1 + w\boldsymbol{M}_2)^{-\frac{1}{2}}$, where

$$\boldsymbol{M}_1 = \boldsymbol{I}_m + (2m^2 + 6m + 5)\boldsymbol{1}_m\boldsymbol{1}_m^\top - (2m + 3)(\boldsymbol{\eta}_m\boldsymbol{1}_m^\top + \boldsymbol{1}_m\boldsymbol{\eta}_m^\top) + 2\boldsymbol{\eta}_m\boldsymbol{\eta}_m^\top,$$

$$\boldsymbol{M}_2 = 2(m^2 + 3m + 2)\mathbf{1}_m\mathbf{1}_m^\top - 2(m+2)(\boldsymbol{\eta}_m\mathbf{1}_m^\top + \mathbf{1}_m\boldsymbol{\eta}_m^\top) + 2\boldsymbol{\eta}_m\boldsymbol{\eta}_m^\top + 2\boldsymbol{H}_m$$

are both positive definite. To see $\boldsymbol{M}_2$ is positive definite, note that $\boldsymbol{M}_2 = 2\boldsymbol{R}\boldsymbol{H}_{m+2}\boldsymbol{R}^\top$ from the proof of Theorem 2. The Cholesky decomposition gives $\boldsymbol{H}_{m+2} = \boldsymbol{C}\boldsymbol{C}^\top$ where $\boldsymbol{C}$ is a lower triangular matrix with $C_{ij} = 1$ for all $i \le j$. It implies $\boldsymbol{H}_{m+2}$ is positive definite. Moreover, $\boldsymbol{R}$ is of full rank as it contains $-\boldsymbol{I}_m$. Overall, this indicates $\boldsymbol{M}_2$ is positive definite. For $\boldsymbol{M}_1$, the proof is similar but easier, so we omit the details here.

Lemma 2 implies that

$$\begin{aligned}
\boldsymbol{\Sigma}_{\rho,\hat{w}}^{-\frac{1}{2}}\boldsymbol{\Sigma}_{\rho,w}^{\frac{1}{2}} &= (\boldsymbol{M}_1 + w\boldsymbol{M}_2 + (\hat{w} - w)\boldsymbol{M}_2)^{-\frac{1}{2}}(\boldsymbol{M}_1 + w\boldsymbol{M}_2)^{-\frac{1}{2}} \\
&= (\boldsymbol{M}_1 + w\boldsymbol{M}_2 + o_P(1))^{-\frac{1}{2}}(\boldsymbol{M}_1 + w\boldsymbol{M}_2)^{-\frac{1}{2}} \\
&= ((\boldsymbol{M}_1 + w\boldsymbol{M}_2)^{-\frac{1}{2}} + o_P(1))(\boldsymbol{M}_1 + w\boldsymbol{M}_2)^{-\frac{1}{2}} \\
&= \boldsymbol{I}_m + o_P(1).
\end{aligned}$$

By Slutsky's theorem, $\boldsymbol{\Sigma}_{\rho,\hat{w}}^{-\frac{1}{2}}\sqrt{n}\hat{\boldsymbol{\rho}} = \left(\boldsymbol{\Sigma}_{\rho,\hat{w}}^{-\frac{1}{2}}\boldsymbol{\Sigma}_{\rho,w}^{\frac{1}{2}}\right)\left(\boldsymbol{\Sigma}_{\rho,w}^{-\frac{1}{2}}\sqrt{n}\hat{\boldsymbol{\rho}}\right) \xrightarrow{D} \mathcal{N}(\mathbf{0}_m, \boldsymbol{I}_m)$, and (26) holds.

**Proof of Proposition 7**: We begin with rewriting the test statistic $n\hat{\boldsymbol{\rho}}^\top\boldsymbol{\Sigma}_{\rho,\hat{w}}^{-1}\hat{\boldsymbol{\rho}}$ as

$$n\hat{\boldsymbol{\gamma}}_m^\top(\hat{\gamma}_0^2\boldsymbol{\Sigma}_{\rho,\hat{w}})^{-1}\hat{\boldsymbol{\gamma}}_m.$$

By an argument similar to that used in the proof of Proposition 6,

$$T_k/(2n) = \gamma_0(1 + kw/2) - \gamma_k + o_p(1), \quad 0 \le k \le m+2.$$

It follows that

$$\begin{aligned}
\hat{\gamma}_0 &= \bar{\mathrm{E}}\hat{\gamma}_0 + o_p(1), \\
\hat{\gamma}_0\hat{w} &= \bar{\mathrm{E}}(\hat{\gamma}_0\hat{w}) + o_p(1), \\
\hat{\gamma}_h &= \gamma_h - (m+2-h)\gamma_{m+1} + (m+1-h)\gamma_{m+2} + o_p(1), \quad 1 \le h \le m;
\end{aligned}$$

and consequently,

$$\hat{\gamma}_0\Sigma_{\rho,\hat{w}} = \bar{\Sigma} + o_p(1).$$

Let $\boldsymbol{\lambda} \in \mathbb{R}^m$ be the vector with $h$-th entry $\gamma_h - (m+2-h)\gamma_{m+1} + (m+1-h)\gamma_{m+2}$ for $1 \le h \le m$. The condition $\mathcal{P}_{m+2}\gamma_{m+2} \ne \mathbf{0}$ implies that $\boldsymbol{\lambda} \ne \mathbf{0}$. Putting the above facts together, we conclude that

$$n\hat{\boldsymbol{\gamma}}_m^\top(\hat{\gamma}_0^2\boldsymbol{\Sigma}_{\rho,\hat{w}})^{-1}\hat{\boldsymbol{\gamma}}_m = n\boldsymbol{\lambda}[(\bar{\mathrm{E}}\hat{\gamma}_0)\bar{\Sigma}]^{-1}\boldsymbol{\lambda}(1 + o_p(1)).$$

Condition 3 guarantees that $w = o(n)$ and hence $\bar{\Sigma} = o(n)$. Therefore, $\lim_{n\to\infty} n\boldsymbol{\lambda}[(\bar{\mathrm{E}}\hat{\gamma}_0)\bar{\Sigma}]^{-1}\boldsymbol{\lambda} = \infty$ and the conclusion of the proposition follows. $\square$

**Proof of Proposition 8**: Direct calculation of the right bottom corner of the matrix leads to

$$1 + [(2m^2 + 6m + 5) + 2(m^2 + 3m + 2)\hat{w}] \cdot 1 - [(2m+3) + 2(m+2)\hat{w}](2m) + (2 + 2\hat{w})m^2 + 2\hat{w}m,$$

which equals to $6 + 4\hat{w}$. $\qquad\qquad\qquad\qquad\qquad\qquad\qquad\qquad\qquad\qquad\qquad\qquad\qquad\qquad$ □

# B   Additional Numerical Results

## B.1   Additional Simulation Studies

### B.1.1   Type I error control

In addition to the empirical results on type I error control in Section 4.1.1, we consider a few more popular tests for autocorrelation besides the classical Box-type tests: (i) Box test calibrated by moving blocks bootstrap (Romano and Thombs, 1996), (ii) Box test calibrated by dependent wild bootstrap (Shao, 2010), and (iii) the Box type test based on distance correlations (Fokianos and Pitsillou, 2017). We name these three tests by MBB, DWB and dCovTS respectively. For all procedures, the lag parameter is set to $m \in \{1, 2, 4, 8\}$. The same data generation processes with three noise distributions as in Section 4.1.1 are used, and the empirical type I error rates, based on 100 replicates, are reported in Table 6. We observe that all these tests fail to control the type I error, with empirical type I error rates equal to 1. Such results are expected as none of these tests is designed for time series with mean shifts.

Table 6: Empirical type I error rates with 100 replicates under three noise distributions: Gaussian (G), Exponential (E) and $t_3$ (T).

| Method | $m = 1$ | $m = 2$ | $m = 4$ | $m = 8$ |
|---|---|---|---|---|
| MMB-G | 1.00 | 1.00 | 1.00 | 1.00 |
| DWB-G | 1.00 | 1.00 | 1.00 | 1.00 |
| dCovTS-G | 1.00 | 1.00 | 1.00 | 1.00 |
| MMB-E | 1.00 | 1.00 | 1.00 | 1.00 |
| DWB-E | 1.00 | 1.00 | 1.00 | 1.00 |
| dCovTS-E | 1.00 | 1.00 | 1.00 | 1.00 |
| MMB-T | 1.00 | 1.00 | 1.00 | 1.00 |
| DWB-T | 1.00 | 1.00 | 1.00 | 1.00 |
| dCovTS-T | 1.00 | 1.00 | 1.00 | 1.00 |

### B.1.2   Power analysis

In this section, we carry out a more comprehensive power analysis with various levels of autocorrelation, different mean structures, a wider range of sample sizes, and additional types of dependence structures. For test methods, we focus on the proposed two SIP variants and the oracle method as a benchmark.

**Various strengths of serial dependence.** To assess the sensitivity of the tests to the strength of serial dependence and sample size, we extend the MA(1) and AR(1) experiments in Section 4.1.2 by considering a finer parameter grid and a broader range of sample sizes. Specifically, the coefficient $\omega$ in MA(1) and $\phi$ in AR(1) vary over a grid $\{-0.2, -0.19, \ldots, 0.19, 0.2\}$ of mesh size 0.01, and we consider $n \in \{2000, 5000, 10000, 20000\}$. For each $n$, the number of mean shifts is set to $J = n/100$. The locations of change points are generated randomly subject to the constraint $\mathcal{L}(\boldsymbol{\theta}) \geq 20$, and the segment means are drawn independently from a

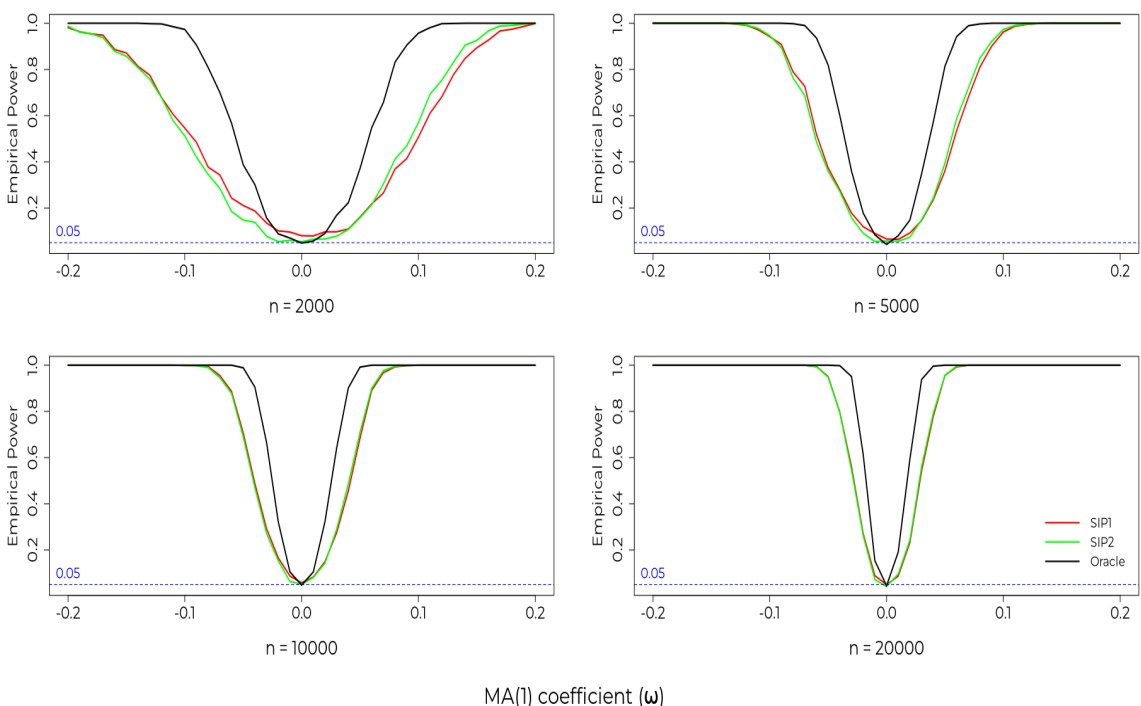

Figure 3: Empirical power curves under MA(1) noise against the coefficient $\omega$, for sample sizes $n = 2000, 5000, 10000,$ and $20000$.

uniform distribution on $[-5, 5]$. For each sample size, the mean vector is generated once and fixed across all parameter settings. Each configuration is replicated 1000 times. The lag parameter $m = 4$ is used for SIP1 and SIP2. The resulting power curves are presented in Figures 3 and 4 for the MA(1) and AR(1) experiments, respectively. We see that the two SIP tests perform similarly and reasonably well. When $n = 2000$, the type I error of SIP1 is slightly above the desired level. Moreover, it seems that the relative power of two SIP tests depends on the sign of autocorrelations.

**Additional dependence patterns.** Besides the scenarios considered in Section 4.1.2, we include two additional time series models here. First, we consider a noise process under the ARMA(1,1) model:

$$\epsilon_t = \phi\,\epsilon_{t-1} + z_t + \omega\,z_{t-1}, \quad z_t \overset{\text{IID}}{\sim} \mathcal{N}(0, 1),$$

where $\phi$ and $\omega$ are the AR and MA coefficients, respectively. We set $\phi = 0.05$ and $\omega = 0.05$ so that the autocorrelations are very weak and consider $n = 2000, 5000, 10000,$ and $20000$. The mean vectors are generated same as before. Table 7 presents the power of the SIP tests and the oracle procedure with the lag parameter $m \in \{1, 2, 4, 8\}$, based on 1000 replicates. We find that the SIP tests have relatively lower powers than the oracle when $n = 2000$, but they gradually catch up as the sample size increases.

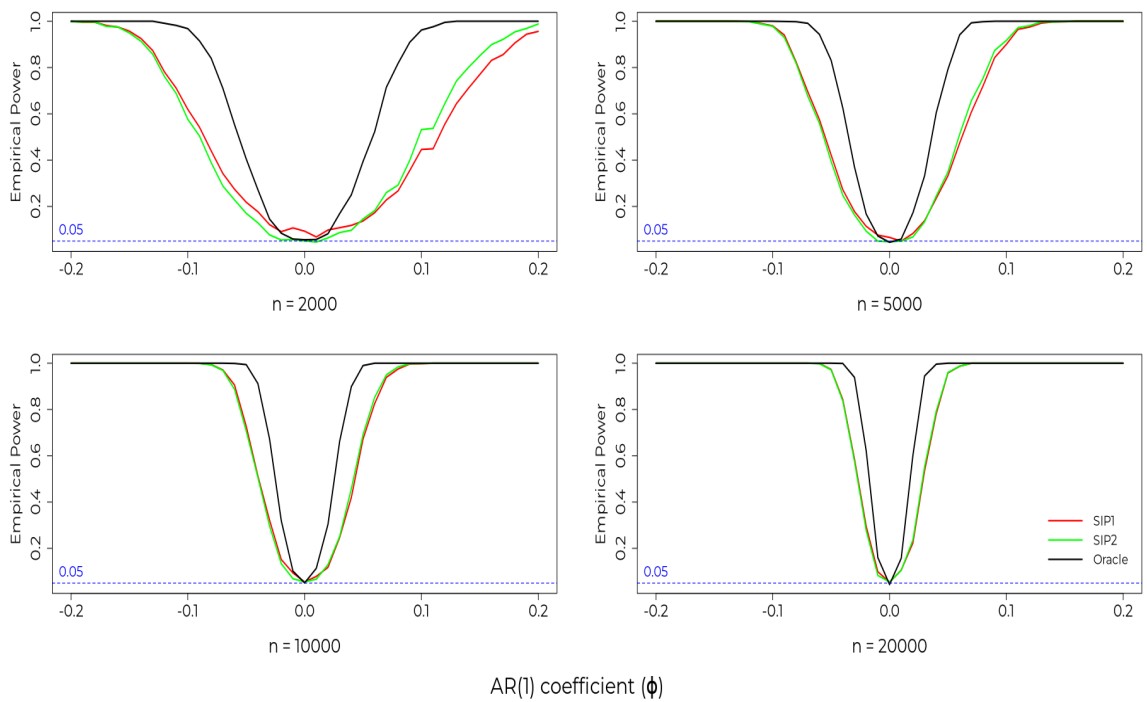

Figure 4: Empirical power curves under AR(1) noise against the coefficient $\phi$, for sample sizes $n = 2000, 5000, 10000$, and $20000$.

Table 7: Empirical power under the ARMA(1,1) noise.

| $n$ | Method | $m = 1$ | $m = 2$ | $m = 4$ | $m = 8$ |
|---|---|---|---|---|---|
| | SIP1 | 0.300 | 0.410 | 0.447 | 0.470 |
| 2000 | SIP2 | 0.386 | 0.489 | 0.521 | 0.526 |
| | Oracle | 0.994 | 0.987 | 0.953 | 0.922 |
| | SIP1 | 0.729 | 0.873 | 0.937 | 0.912 |
| 5000 | SIP2 | 0.777 | 0.896 | 0.952 | 0.955 |
| | Oracle | 1.000 | 1.000 | 1.000 | 1.000 |
| | SIP1 | 0.959 | 0.997 | 0.998 | 1.000 |
| 10000 | SIP2 | 0.965 | 0.999 | 0.998 | 1.000 |
| | Oracle | 1.000 | 1.000 | 1.000 | 1.000 |
| | SIP1 | 1.000 | 1.000 | 1.000 | 1.000 |
| 20000 | SIP2 | 1.000 | 1.000 | 1.000 | 1.000 |
| | Oracle | 1.000 | 1.000 | 1.000 | 1.000 |

In the second experiment, we consider a nonlinear time series model

$$\epsilon_t = (a + bz_t)\varepsilon_{t-1} + z_t, \quad z_t \overset{\text{IID}}{\sim} N(0,1).$$

This is a bilinear process whose lag-$h$ autocorrelation is $a^h$, and we set $a = 0.1$, and $b = 0.5$ so that the autocorrelations are weak. The results are recorded in Table 8. Patterns similar to

those in Table 7 are observed again.

Table 8: Empirical power under the bilinear noise.

| $n$ | Method | $m = 1$ | $m = 2$ | $m = 4$ | $m = 8$ |
|---|---|---|---|---|---|
| | SIP1 | 0.213 | 0.339 | 0.448 | 0.436 |
| 2000 | SIP2 | 0.316 | 0.439 | 0.510 | 0.487 |
| | Oracle | 0.959 | 0.947 | 0.889 | 0.819 |
| | SIP1 | 0.703 | 0.864 | 0.931 | 0.912 |
| 5000 | SIP2 | 0.749 | 0.889 | 0.945 | 0.941 |
| | Oracle | 0.999 | 0.999 | 0.997 | 0.994 |
| | SIP1 | 0.945 | 0.995 | 0.999 | 1.000 |
| 10000 | SIP2 | 0.959 | 0.997 | 0.999 | 1.000 |
| | Oracle | 1.000 | 1.000 | 1.000 | 1.000 |
| | SIP1 | 1.000 | 1.000 | 1.000 | 1.000 |
| 20000 | SIP2 | 1.000 | 1.000 | 1.000 | 1.000 |
| | Oracle | 1.000 | 1.000 | 1.000 | 1.000 |

**Various mean structures.** In this experiment we investigate the robustness of the proposed test procedures under mean structures with varying frequencies and magnitudes of mean shifts. We consider the MA(1) and AR(1) models studied previously, with coefficients ($\omega$ and $\phi$, respectively) fixed at 0.1 and sample size $n = 10000$.

The number of mean shifts is set to $J \in \{50, 100, 200\}$, subject to the constraint $\mathcal{L}(\boldsymbol{\theta}) \geq 20$. The segment means are independently generated from uniform distributions over $[-5, 5]$, $[-10, 10]$, and $[-20, 20]$, resulting in a total of nine different mean configurations for each model. The results, based on 1000 replicates, are summarized in Tables 9 and 10.

Consistent with our theoretical findings, the power of the SIP tests decreases as the total variation of the mean increases. In the most extreme setting, where the mean varies over $[-20, 20]$, the performance deterioration becomes more pronounced. However, this regime corresponds to a relatively simple scenario in which most change points can be reliably detected and removed using standard methods. In such cases, we do not recommend applying the SIP tests directly.

Table 9: Average power under MA(1) noise with varying shift amplitude and frequency with 1000 replicates.

| Segment Mean | Segments | Method | $m = 1$ | $m = 2$ | $m = 4$ | $m = 8$ |
|---|---|---|---|---|---|---|
| | | SIP1 | 0.995 | 1.000 | 1.000 | 1.000 |
| | 50 | SIP2 | 0.996 | 1.000 | 1.000 | 1.000 |
| | | Oracle | 1.000 | 1.000 | 1.000 | 1.000 |
| $[-5, 5]$ | | SIP1 | 0.988 | 0.998 | 1.000 | 1.000 |
| | 100 | SIP2 | 0.991 | 0.999 | 1.000 | 1.000 |
| | | Oracle | 1.000 | 1.000 | 1.000 | 1.000 |
| | | SIP1 | 0.974 | 0.997 | 0.999 | 0.997 |
| | 200 | SIP2 | 0.979 | 0.999 | 0.999 | 1.000 |
| | | Oracle | 1.000 | 1.000 | 1.000 | 1.000 |
| | | SIP1 | 0.966 | 0.993 | 0.998 | 0.995 |
| | 50 | SIP2 | 0.969 | 0.995 | 0.998 | 0.994 |
| | | Oracle | 1.000 | 1.000 | 1.000 | 1.000 |
| $[-10, 10]$ | | SIP1 | 0.944 | 0.979 | 0.987 | 0.968 |
| | 100 | SIP2 | 0.953 | 0.984 | 0.990 | 0.981 |
| | | Oracle | 1.000 | 1.000 | 1.000 | 1.000 |
| | | SIP1 | 0.876 | 0.939 | 0.944 | 0.888 |
| | 200 | SIP2 | 0.902 | 0.948 | 0.954 | 0.921 |
| | | Oracle | 1.000 | 1.000 | 1.000 | 1.000 |
| | | SIP1 | 0.884 | 0.942 | 0.959 | 0.907 |
| | 50 | SIP2 | 0.902 | 0.950 | 0.975 | 0.946 |
| | | Oracle | 1.000 | 1.000 | 1.000 | 1.000 |
| $[-20, 20]$ | | SIP1 | 0.717 | 0.784 | 0.785 | 0.676 |
| | 100 | SIP2 | 0.758 | 0.817 | 0.821 | 0.752 |
| | | Oracle | 1.000 | 1.000 | 1.000 | 1.000 |
| | | SIP1 | 0.434 | 0.472 | 0.442 | 0.366 |
| | 200 | SIP2 | 0.512 | 0.536 | 0.494 | 0.393 |
| | | Oracle | 1.000 | 1.000 | 1.000 | 1.000 |

Table 10: Average power under AR(1) noise with varying shift amplitude and frequency with 1000 replicates.

| Segment Mean | Segments | Method | $m = 1$ | $m = 2$ | $m = 4$ | $m = 8$ |
|---|---|---|---|---|---|---|
| | | SIP1 | 0.937 | 0.994 | 0.999 | 1.000 |
| | 50 | SIP2 | 0.948 | 0.997 | 1.000 | 1.000 |
| | | Oracle | 1.000 | 1.000 | 1.000 | 1.000 |
| | | SIP1 | 0.928 | 0.990 | 0.997 | 0.998 |
| $[-5, 5]$ | 100 | SIP2 | 0.937 | 0.993 | 0.998 | 0.999 |
| | | Oracle | 1.000 | 1.000 | 1.000 | 1.000 |
| | | SIP1 | 0.903 | 0.981 | 0.998 | 0.991 |
| | 200 | SIP2 | 0.917 | 0.987 | 0.999 | 0.999 |
| | | Oracle | 1.000 | 1.000 | 1.000 | 1.000 |
| | | SIP1 | 0.910 | 0.974 | 0.996 | 0.995 |
| | 50 | SIP2 | 0.929 | 0.982 | 0.999 | 0.999 |
| | | Oracle | 1.000 | 1.000 | 1.000 | 1.000 |
| | | SIP1 | 0.820 | 0.934 | 0.977 | 0.954 |
| $[-10, 10]$ | 100 | SIP2 | 0.845 | 0.946 | 0.981 | 0.976 |
| | | Oracle | 1.000 | 1.000 | 1.000 | 1.000 |
| | | SIP1 | 0.729 | 0.834 | 0.877 | 0.818 |
| | 200 | SIP2 | 0.762 | 0.871 | 0.882 | 0.864 |
| | | Oracle | 1.000 | 1.000 | 1.000 | 1.000 |
| | | SIP1 | 0.764 | 0.876 | 0.899 | 0.866 |
| | 50 | SIP2 | 0.793 | 0.892 | 0.918 | 0.907 |
| | | Oracle | 1.000 | 1.000 | 1.000 | 1.000 |
| | | SIP1 | 0.520 | 0.587 | 0.587 | 0.537 |
| $[-20, 20]$ | 100 | SIP2 | 0.573 | 0.647 | 0.644 | 0.593 |
| | | Oracle | 1.000 | 1.000 | 1.000 | 1.000 |
| | | SIP1 | 0.336 | 0.388 | 0.384 | 0.383 |
| | 200 | SIP2 | 0.402 | 0.445 | 0.445 | 0.382 |
| | | Oracle | 1.000 | 1.000 | 1.000 | 1.000 |

## B.2   Additional Real Data Examples

### B.2.1   HadCET dataset

The Hadley Centre Central England Temperature (HadCET) dataset (Parker et al., 1992) records yearly averages of the monthly mean, maximum, and minimum temperatures over a representative region in the United Kingdom. We follow the change-point analysis in Cho and Fryzlewicz (2024) and focus on the period 1878–2019 ($n = 142$) for these three time series.

This dataset represents a classical time series with occasional structural changes in the mean. Although mean shifts in this dataset are relatively infrequent and can be detected using modern change-point methods, it provides a useful benchmark for illustrating how unaccounted mean shifts may induce spurious temporal dependence. Indeed, Cho and Fryzlewicz (2024) identifies

two change points in each of the three series. Motivated by this, we consider both the original series and the demeaned series obtained after change-point adjustment. Table 11 reports the $p$-values of the proposed SIP1 and SIP2 tests, along with the classical Box test applied to the original data and to the demeaned data. For all methods, the lag order is set to $m \in \{1, 2, 4, 8\}$.

The Box test applied to the original series suggests significant temporal dependence in all three series. In contrast, the SIP tests yield consistently moderate to large $p$-values, indicating little evidence of dependence. Notably, the Box test applied to the demeaned series produces results broadly consistent with the SIP tests.

This comparison suggests that the apparent autocorrelation detected by the classical Box test is largely due to mean shifts rather than genuine serial dependence in the noise. The SIP results align with analyses that explicitly remove change points prior to autocorrelation estimation, reinforcing that the underlying noise structure is close to uncorrelated. Figure 5 displays the time series along with the standard ACF and the shift-immune ACF. The standard ACF suggests persistent autocorrelation across multiple lags, whereas the shift-immune ACF remains largely clean, indicating little evidence of intrinsic dependence after accounting for mean shifts.

Table 11: Summary of test results on HadCET data across lags.

| data | test | $m = 1$ | $m = 2$ | $m = 4$ | $m = 8$ |
|---|---|---|---|---|---|
| Mean | SIP1 | 0.452 | 0.692 | 0.144 | 0.668 |
| | SIP2 | 0.519 | 0.492 | 0.175 | 0.314 |
| | Box (original) | $5.95 \times 10^{-8}$ | $3.82 \times 10^{-12}$ | $4.11 \times 10^{-15}$ | $< 2.2 \times 10^{-16}$ |
| | Box (demeaned) | 0.429 | 0.500 | 0.173 | 0.551 |
| Max | SIP1 | 0.592 | 0.615 | 0.231 | 0.503 |
| | SIP2 | 0.632 | 0.351 | 0.132 | 0.302 |
| | Box (original) | $1.86 \times 10^{-7}$ | $5.21 \times 10^{-11}$ | $2.33 \times 10^{-13}$ | $< 2.2 \times 10^{-16}$ |
| | Box (demeaned) | 0.369 | 0.484 | 0.250 | 0.699 |
| Min | SIP1 | 0.308 | 0.749 | 0.449 | 0.962 |
| | SIP2 | 0.405 | 0.637 | 0.462 | 0.573 |
| | Box (original) | $1.47 \times 10^{-6}$ | $3.14 \times 10^{-10}$ | $1.04 \times 10^{-12}$ | $< 2.2 \times 10^{-16}$ |
| | Box (demeaned) | 0.707 | 0.622 | 0.351 | 0.683 |

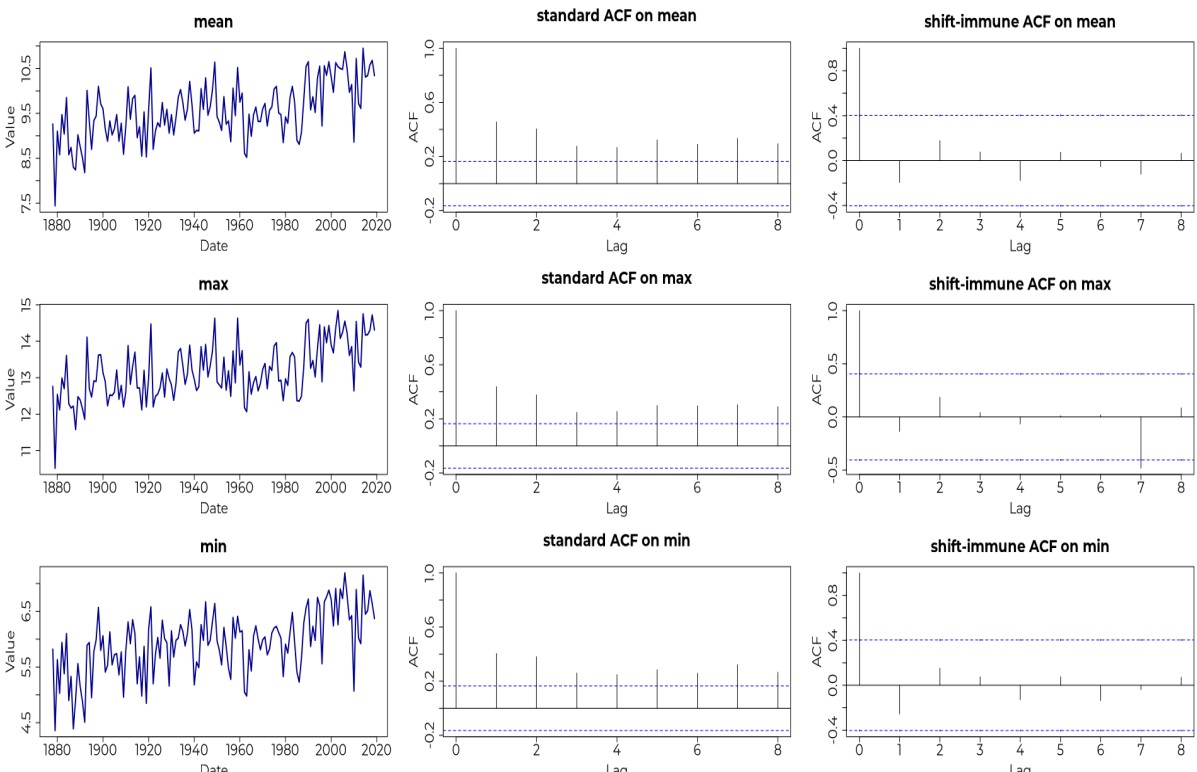

Figure 5: Left: yearly average of the mean, maximum and minimum monthly temperatures (top to bottom). Middle and right: standard ACF plots and shift-immune ACF plots of corresponding data.

### B.2.2 Pulsar dataset

Pulsar timing residuals measure the deviations between the observed arrival times of pulses from a pulsar, an ultra-dense remnant of an exploded massive star, and the arrival times predicted by a precise timing model. As noted in D'Alessandro et al. (1995), these residuals are often modeled as piecewise-constant mean segments, where abrupt shifts (associated with glitches or microglitches) correspond to change points in the mean and reflect rapid changes in the pulsar's spin behavior.

The noise structure of pulsar timing residuals are approximately white when the pulsar is intrinsically stable and the timing model accounts for all dominant deterministic effects. In contrast, autocorrelation arises when slow physical or environmental processes introduce long-term memory (Hobbs et al., 2010; Shannon and Cordes, 2010). Consequently, it is believed that pulsars such as PSR J1713+0747 exhibit exceptionally clean, near-white residuals, whereas PSR J1643−1224 displays strong low-frequency (red) timing noise (Arzoumanian et al., 2015; Lam et al., 2017). We analyze the data on these two pulsars obtained from the NANOGrav Public Data Portal (https://nanograv.org/science/data/11-year-pulsar-timing-array-data-release).

Table 12 reports the $p$-values of the Box test and SIP tests applied to the pulsar timing residuals of PSR J1713+0747 ($n = 27570$) and PSR J1643−1224 ($n = 11527$). While the Box test suggests significant autocorrelation for both pulsars, the results from SIP tests (with lag choices $m = 2$ or 4) align with the expected behavior, i.e., the residuals of PSR J1713+0747 are white noise, whereas those of PSR J1643−1224 exhibit red noise. We highlight in bold a few $p$-values inconsistent with the expected behavior. Specifically, SIP1 with $m = 8$ leads

to a marginally small $p$-value for PSR J1713+0747. Both SIP tests with $m = 1$ fail to show significance for PSR J1643−1224, which aligns well with the shift-immune ACF plot. With larger $m = 2, 4$, both tests lead to conclusions consistent with the expected behavior.

Overall, this example demonstrates that the proposed method not only improves robustness to mean shifts, but also preserves scientifically meaningful distinctions that are obscured by classical approaches. Figure 6 displays the raw data along with the standard ACF and the shift-immune ACF for both pulsars, illustrating the contrast between the two methods.

Table 12: Summary of test results on pulsar timing residuals across lags.

| object | test | $m = 1$ | $m = 2$ | $m = 4$ | $m = 8$ |
|---|---|---|---|---|---|
| | SIP1 | 0.086 | 0.145 | 0.225 | 0.192 |
| PSR J1713+0747 | SIP2 | 0.078 | 0.141 | 0.217 | **0.024** |
| (white noise) | Box | $< 2.2 \times 10^{-16}$ | $< 2.2 \times 10^{-16}$ | $< 2.2 \times 10^{-16}$ | $< 2.2 \times 10^{-16}$ |
| | SIP1 | **0.888** | 0.026 | 0.017 | **0.081** |
| PSR J1643−1224 | SIP2 | **0.888** | 0.010 | 0.008 | 0.033 |
| (red noise) | Box | $< 2.2 \times 10^{-16}$ | $< 2.2 \times 10^{-16}$ | $< 2.2 \times 10^{-16}$ | $< 2.2 \times 10^{-16}$ |

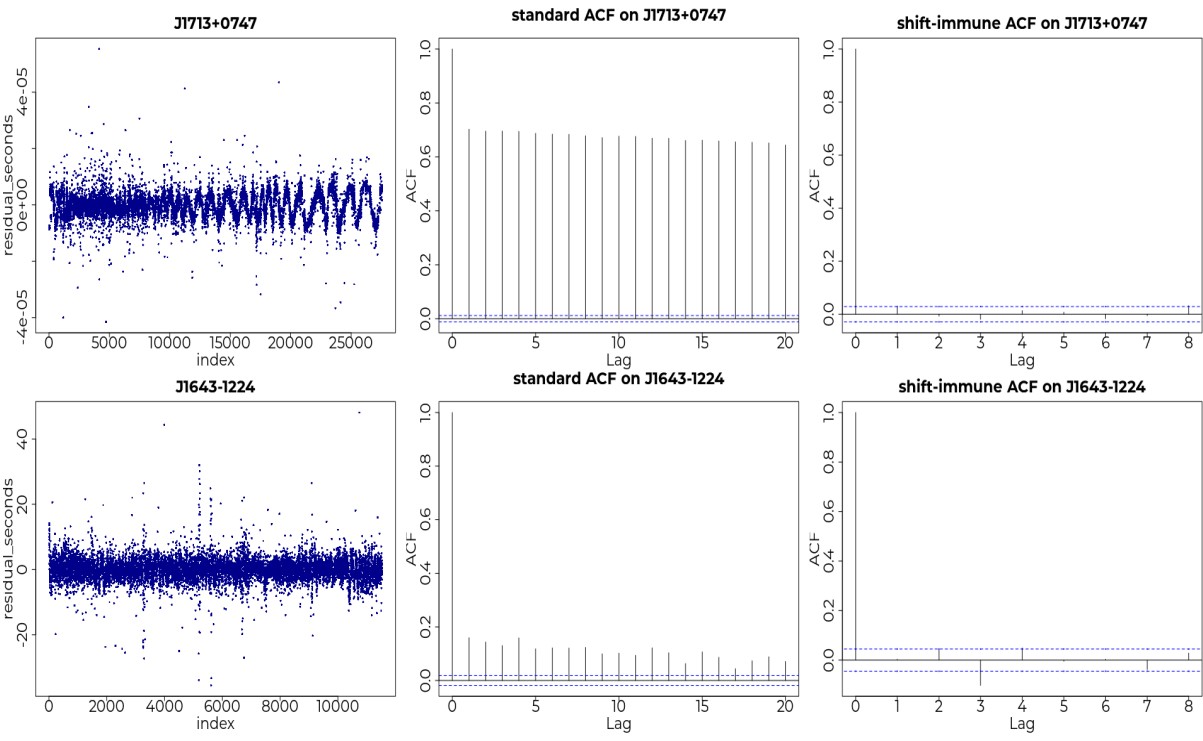

Figure 6: Left: pulsar timing residuals. Middle and right: standard ACF plots and shift-immune ACF plots of corresponding data.

Taken together, these examples reveal a consistent pattern across different application domains. In the presence of mean shifts for large datasets, classical portmanteau tests tend to overstate evidence of temporal dependence by confounding structural changes with autocorrelation. In contrast, the proposed SIP tests effectively separate intrinsic noise dependence from such artifacts, yielding conclusions that align with domain-specific scientific understanding. This robustness makes the SIP approach particularly well suited for modern time series applications,

where structural changes and large-scale data are ubiquitous.

## Code Availability

An R package `SIP` and all analysis code are available in the GitHub repository: [https://github.com/ziyang773/SIP](https://github.com/ziyang773/SIP).

