# OpenReview forum: "Autocorrelation Test under Frequent Mean Shifts"
_SLADS/Section_B — Accepted by SLADS_Section_B_

### Review · Reviewer_gCrv · 2026-02-24

**Summary Of Contributions:**

This submission studies \textbf{autocorrelation testing for time series with frequent mean shifts}, a setting in which classical Box-type portmanteau tests can fail badly because nonstationary piecewise-constant means induce spurious autocorrelation. The paper proposes a \textbf{Shift-Immune Portmanteau (SIP)} framework that is designed to be robust to such mean shifts and is motivated in part by nanopore sequencing signals.

The core methodological contribution is a characterization of a class of quadratic forms $X^\top A X$ (with symmetric Toeplitz / circulant structure) whose expectations eliminate the effect of the piecewise-constant mean under a minimum segment-length condition. In particular, the paper derives necessary and sufficient algebraic conditions on $A$ (Theorem 1), then adds global-shift invariance and variance-removal constraints to define a usable class of test-building quadratic forms.

Using these building blocks, the authors construct the SIP test statistic as a properly standardized quadratic form of transformed lag-difference statistics $T_h$, and show that the construction is invariant to the choice of basis for the relevant subspace. The paper further clarifies what is being tested by identifying the projection $P_{m+2}\gamma_{m+2}$, and discusses when asymptotic power is expected.

On the theory side, the paper establishes asymptotic normality for the SIP autocovariance estimators and an asymptotic $\chi_m^2$ null distribution for the SIP statistic (Theorems 2--3) under stated conditions, including an IID null and segment-length constraints involving $L$ and $m$, and also provides a stochastic-dominance statement under relaxed conditions.

Practically, the paper proposes two versions (SIP 1 / SIP 2), a shift-immune ACF-style visualization, simulation studies (type I error and power), and a nanopore sequencing application. The simulations show severe failure of Box-Pierce under frequent mean shifts and strong null calibration of SIP, especially SIP 2; the real-data analysis reports significant serial correlation across the nanopore sequences studied.

**Audience:**

Yes

**Claims And Evidence:**

Yes

**Requested Changes:**

Add some simulations and theoretical results.

**Strengths And Weaknesses:**

Strengths: The paper addresses a real gap: testing serial correlation before (or without) reliable mean-shift estimation in highly nonstationary signals (e.g., nanopore sequencing). This is a strong and timely problem formulation. The SIP-ACF visualization broadens the contribution beyond hypothesis testing and improves usability for practitioners working with nonstationary signals.The paper includes detailed derivations and asymptotic arguments, which strengthens rigor and reproducibility.

Weaknesses:
1. Limited benchmarking in simulations and real data analysis.
The simulation studies and real-data application compare the proposed method only with the classical Box–Pierce test. This comparison is too limited. Additional competing methods (especially modern alternatives) should be included to provide a more comprehensive and convincing evaluation.

2. Lack of power analysis.
The manuscript does not provide a sufficient power analysis. A systematic investigation of power under different alternatives (e.g., varying signal strength, sample size, and dependence structures) is needed to better demonstrate the advantages and limitations of the proposed method.

3. Insufficient theoretical explanation for the failure of the Box–Pierce test.
The paper mentions that the Box–Pierce test performs poorly, but it does not provide theoretical insight into why this happens. Adding theoretical results (or at least a clear analytical explanation) on the failure mechanism of the Box–Pierce test would strengthen the manuscript significantly.

---

> ### Author Response · Authors · 2026-04-13
> **Response to Reviewer gCrv**
>
> We would like to thank the reviewer for the helpful and constructive comments. We greatly appreciate the positive assessment on our problem formulation and methodological contributions, as well as the helpful suggestions on benchmarking, power analysis, and theoretical clarification. These comments have been very valuable for our revision.
>
> We have substantially strengthened the paper. In particular, we expand the simulations to include additional methods, conduct a more comprehensive power analysis across different dependence strengths, sample sizes, and mean-shift configurations, and clarify the mechanism underlying the failure of classical Box-type tests. These revisions improve both the empirical validation and the clarity of the manuscript.
>
> We highlighted all changes in blue (for the main text). In addition, the entire section of appendix B is new. Please see our point-to-point responses below.
>
>
>
> > 1. Limited benchmarking in simulations and real data analysis. The simulation studies and real-data application compare the proposed method only with the classical Box–Pierce test. This comparison is too limited. Additional competing methods (especially modern alternatives) should be included to provide a more comprehensive and convincing evaluation.
>
> **Response:**
> We have substantially expanded the benchmarking study in Appendix B. In addition to the classical Box test, we now include several widely used alternatives, including the moving blocks bootstrap (Romano and Thombs, 1996), the dependent wild bootstrap (Shao, 2010),  and a distance-correlation-based test (Fokianos and Pitsillou, 2017).
>
> Our results in Appendix B.1 show that these methods also fail to control type I error in the presence of mean shifts, further highlighting the necessity of the proposed SIP framework.
>
> We note that these methods are primarily designed for stationary or weakly dependent time series without structural changes in the mean. Therefore, their performance limitations in this setting are expected. To the best of our knowledge, the SIP framework is among the first approaches specifically designed for testing autocorrelation in time series with frequent mean shifts, and there are currently limited alternative methods tailored to this setting.
>
> > 2. Lack of power analysis. The manuscript does not provide a sufficient power analysis. A systematic investigation of power under different alternatives (e.g., varying signal strength, sample size, and dependence structures) is needed to better demonstrate the advantages and limitations of the proposed method.
>
> **Response:**
> We have expanded the power analysis to cover a broader range of scenarios in Appendix B.2. Specifically, we consider multiple sample sizes (from 2,000 to 20,000), additional dependence structures (including ARMA and bilinear models), finer grids of dependence parameters to capture varying signal strengths, and a variety of mean shift configurations with different frequencies and magnitudes.
>
> These additions provide a more comprehensive assessment of the power of the proposed methods across diverse settings.
>
>
> > 3 Insufficient theoretical explanation for the failure of the Box–Pierce test. The paper mentions that the Box–Pierce test performs poorly, but it does not provide theoretical insight into why this happens. Adding theoretical results (or at least a clear analytical explanation) on the failure mechanism of the Box–Pierce test would strengthen the manuscript significantly.
>
>
> **Response:**
> Originally, we provided an intuitive explanation of the failure of the classical Box test under frequent mean shifts by pointing out that the expectations of the sample autocovariances depend on the total variation of the mean function. Following your suggestion, we have augmented this discussion at the beginning of Section 2.2. In particular, equation (4) shows that the Box statistic involves both $\gamma_h$ and $\check{\gamma}_h(\theta)$, which can lead to spurious autocorrelation when the mean is non-constant.

---

> > ### Comment · Reviewer_gCrv · 2026-04-22
> >
> > Thank you very much to the authors for their careful response to my previous comments. The revised manuscript includes additional simulation comparisons with other methods and a power analysis, both of which substantially enhance the contribution and completeness of the paper. Overall, I am very satisfied with the revisions and have no further concerns. I recommend acceptance.

---

### Review · Reviewer_G6sv · 2026-03-24

**Summary Of Contributions:**

This paper primarily investigates the problem of testing for autocorrelation in nonstationary time series with frequent mean shifts. The main contributions are as follows: it proposes a novel autocorrelation testing framework that does not rely on change-point detection—the Shift-Immune Portmanteau (SIP) test; it establishes the asymptotic distribution of the test statistic and proves that, under the null hypothesis, the statistic follows a $\chi^2$ distribution; it introduces a new visualization tool for dependence structures in nonstationary time series—the Shift-Immune ACF plot; and it provides empirical evidence supporting the presence of significant positive autocorrelation in nanopore sequencing data, thereby advancing research in this area.

**Audience:**

Yes

**Broader Impact Concerns:**

I do not see a major ethical concern that would require a substantial Broader Impact Statement. The paper is primarily methodological and focuses on statistical inference for nonstationary time series. The nanopore sequencing application does not, in itself, raise an immediate ethical issue based on the material currently presented.

**Claims And Evidence:**

Yes

**Requested Changes:**

1. The paper primarily establishes the asymptotic theory under the null hypothesis $H_0$ and discusses conditions under which the test has asymptotic power. However, it does not derive the limiting distribution under fixed or local alternatives. As a result, it is difficult to rigorously characterize the test’s discriminative ability, detection boundary, and theoretical power under varying degrees of deviation from the null. Although the paper notes that when $P_{m+2} \gamma_{m+2}\ne0$, “the test must have asymptotic power for some m,” this remains a qualitative statement rather than a complete theoretical treatment under the alternative hypothesis.
2. The current power experiments primarily vary $\omega$ and $\phi$ in MA(1), MA(4), and AR(1) models, while the mean shift structure remains largely fixed. This is insufficient to fully support the central claim of robustness to frequent mean shifts. In particular, the experiments do not systematically disentangle the effects of mean shift characteristics (e.g., jump magnitude and change-point frequency) from those of the dependence structure (e.g., correlation strength and form) on test power. It is recommended to vary the jump size, the minimum segment length $L$, and the frequency of change points. Additionally, a finer parameter grid for $\omega$ and $\phi$ near zero should be considered to examine the sensitivity of power to these factors, especially under weak dependence.
3. The implementation of the method relies on both $m$ and $L$: the theory requires $m+2\le L$ or $m+2\le L/2$, which is described in the paper as a “reasonable assumption.” However, this justification remains insufficient, particularly because L is typically unknown in practice. Moreover, the simulations fix $m\in \{1,2,4,8\} $, while the real data analysis recommends $m=4$  as a default choice. This appears to be more of a heuristic guideline rather than a data-driven selection. It is therefore suggested that the authors provide a more principled rule for choosing m, propose feasible strategies when $L$ is unknown, and discuss the potential impact when the stated condition is violated. In addition, robustness analyses with respect to different choices of $m$ should be included in the real data application.
4. The paper ultimately favors SIP 2, yet it does not sufficiently explain why the two statistics, $T_{n,1}^{SIP}$  and $T_{n,2}^{SIP}$, exhibit different finite-sample performance, nor does it clarify their respective domains of applicability or the sources of their differences in stability. A more in-depth analysis is warranted. In particular, the authors should more explicitly articulate the differences in the construction of the two statistics and their corresponding statistical interpretations, rather than relying primarily on empirical comparisons.
5. Section 3.3 would benefit from a clearer definition of the testing target. The overarching objective of the paper is to test whether the process is free of serial correlation, i.e., $\rho_h=0$ for all $h>0$. However, for a given m, the formal hypothesis actually tested in the paper is $P_{m+2} \gamma_{m+2}=0$. It is recommended that the authors explicitly distinguish, in Section 3.3, between the overall research objective and the formal statistical hypothesis under a fixed m, and clarify the relationship between the two. This would help avoid potential confusion where readers may directly equate “testing all autocorrelations being zero” with “testing $P_{m+2} \gamma_{m+2}=0$.”
6. The manuscript still contains several noticeable typographical errors and grammatical issues, and a thorough proofreading is recommended to ensure consistency and clarity throughout. For example:

$\qquad$(1) in “In this subsection. we investigate ...”, the punctuation is incorrect;

$\qquad$(2) in “if m an integer such that ...”, a linking verb is missing;

$\qquad$(3) there are occasional spelling errors in table captions and the main text.

**Strengths And Weaknesses:**

**Strengths**

1. The paper addresses a genuinely difficult setting that standard autocorrelation tests do not handle well. The focus is on nonstationary time series with frequent mean shifts, where classical tests can be distorted by nonconstant means, and change-point-first procedures may be unreliable when segmentation is unstable.

2. The SIP framework gives a direct methodological response to this limitation. Instead of relying on prior change-point estimation, the paper constructs a shift-immune test from transformed autocovariance quantities, offering a technically distinct alternative to existing approaches.

3. The paper contributes both a testing procedure and a diagnostic tool. Besides the SIP test, it introduces the Shift-Immune ACF plot, which improves the practical interpretability of the framework.

4. The empirical application matches the methodological motivation. The nanopore sequencing example is a plausible setting where mean instability and serial dependence may coexist, so the application is relevant rather than merely illustrative.

**Weaknesses**

1. The theory under alternatives remains incomplete relative to the paper’s inferential goal. The paper rigorously derives the null asymptotic distribution, but its treatment under alternatives is mainly qualitative. This limits a precise theoretical assessment of discriminative ability and power.

2. The simulations do not yet isolate the source of the claimed robustness. The power study mainly varies dependence parameters in MA(1), MA(4), and AR(1) models, while the mean-shift structure is largely fixed. It therefore remains unclear whether the reported robustness is driven by jump size, change-point frequency, segment length, or only by the dependence settings considered.

---

> ### Author Response · Authors · 2026-04-13
> **Response to Reviewer G6sv (part 1)**
>
> We sincerely thank the reviewer for the careful reading and constructive comments. We appreciate the positive assessment of our problem formulation, methodological contributions, and practical relevance, as well as the helpful suggestions on theory, simulation design, and presentation. These comments have been helpful in improving the manuscript.
>
> In response, we have revised the paper accordingly. We expand the simulation studies to systematically vary mean-shift characteristics, dependence strength, and sample size, providing a clearer assessment of power. We also clarify the behavior under alternatives, better explain the differences between SIP1 and SIP2, and revise Section 3.3 to more clearly define the testing target. Finally, we have carefully proofread the manuscript to address typographical and grammatical issues.
>
> We highlighted all changes in blue (for the main text). In addition, the entire section of appendix B is new. Please see our point-to-point responses below.
>
> > 1. The paper primarily establishes the asymptotic theory under the null hypothesis $H_0$ and discusses conditions under which the test has asymptotic power. However, it does not derive the limiting distribution under fixed or local alternatives. As a result, it is difficult to rigorously characterize the test’s discriminative ability, detection boundary, and theoretical power under varying degrees of deviation from the null. Although the paper notes that when $P_{m+2}\gamma_{m+2}$, “the test must have asymptotic power for some $m$,” this remains a qualitative statement rather than a complete theoretical treatment under the alternative hypothesis.
>
> **Response:**
> We agree that a full characterization of the limiting distribution under general alternatives is an important and challenging problem. While deriving such results is beyond the scope of the current work, we have added in Section 3.4 a new Proposition 7 regarding the power of the proposed tests.
> In particular, we show that the test is driven by the projected quantity $P_{m+2}\boldsymbol{\gamma}_{m+2}$. Under alternatives where this projection is nonzero, the test statistic diverges with the sample size, implying asymptotic power 1.
>
> We believe this addition provides a clear conceptual understanding of the discriminative ability of the test without substantially increasing the technical complexity of the paper.
>
> > 2. The current power experiments primarily vary $\omega$ and $\phi$ in MA(1), MA(4), and AR(1) models, while the mean shift structure remains largely fixed. This is insufficient to fully support the central claim of robustness to frequent mean shifts. In particular, the experiments do not systematically disentangle the effects of mean shift characteristics (e.g., jump magnitude and change-point frequency) from those of the dependence structure (e.g., correlation strength and form) on test power. It is recommended to vary the jump size, the minimum segment length $L$, and the frequency of change points. Additionally, a finer parameter grid for $\omega$ and $\phi$ near zero should be considered to examine the sensitivity of power to these factors, especially under weak dependence.
>
> **Response:**
>
> We have substantially expanded the simulation studies in Appendix B to address this concern. In particular, we now
>
> (i) Vary the magnitude of mean shifts (uniform distributions over $[-5,5]$, $[-10,10]$, and $[-20,20])$;
> (ii) Vary the frequency of change points (different values of $J$);
> (iii) include more mean structures (in each scenario, the mean vector is randomly generated based on the frequency and magnitude setup);
> (iv) Consider a finer grid of dependence parameters (the parameters $\omega$ and $\phi$ for MA(1) and AR(1) models, respectively, take values in a fine grid $\{-0.2,-0.19,...,0.2\}$);
> (v) Include a range of sample sizes ($n = 2000, 5000, 10000, 20000)$).
>
> These additions allow us to disentangle the effects of mean structure and dependence strength on test performance. The results confirm that the proposed SIP tests are robust across a wide range of mean shift configurations.

---

> ### Author Response · Authors · 2026-04-13
> **Response to Reviewer G6sv (part 2)**
>
> > 3. The implementation of the method relies on both $m$ and $L$: the theory requires $m+2\leq L$ or $m+2\leq L/2$, which is described in the paper as a “reasonable assumption.” However, this justification remains insufficient, particularly because $L$ is typically unknown in practice. Moreover, the simulations fix $m\in\{1,2,4,8\}$, while the real data analysis recommends $m=4$ as a default choice. This appears to be more of a heuristic guideline rather than a data-driven selection. It is therefore suggested that the authors provide a more principled rule for choosing $m$, propose feasible strategies when $L$ is unknown, and discuss the potential impact when the stated condition is violated. In addition, robustness analyses with respect to different choices of $m$ should be included in the real data application.
>
>
> **Response:**
> We thank the reviewer for this insightful comment. We agree that the choice of the lag parameter $m$ is an important issue, particularly since the minimal segment length $L$ is typically unknown in practice.
>
> In the revision, we include results of the SIP tests with $m \in \{1,2,4,8\}$ in the nanopore data analysis (Section 4.2), along with a couple pf additional real data examples (Appendix B.2). Across these examples, we observe that the conclusions are largely consistent across different choices of $m$. For instance, in the nanopore data analysis, the test results agree across all considered values of $m$ for the vast majority of sequences, with only a small fraction (32 out of 900) showing discrepancies when $m=1$; see the updated Table~5. These findings suggest that the proposed SIP tests are reasonably robust to the choice of $m$ in practice. Based on both simulation and real data results, we recommend $m=4$ as a default choice.
>
> We agree that it would be desirable to develop a fully data-driven or principled rule for selecting $m$. However, even in the classical stationary setting without mean shifts, there is no universally optimal or fully data-driven procedure for choosing $m$ in Box-type portmanteau tests. In practice, $m$ is typically selected based on experience or through examining multiple values. In the nonstationary setting with mean shifts, the problem becomes even more challenging due to the additional complexity introduced by change points.
>
> Regarding the condition $m+2 \leq L$, we emphasize that it is sufficient for the theoretical guarantees, but not strictly necessary for good empirical performance. In the literature of change-point analysis, the minimal segment length $L$ is assumed to grow with the sample size, under which moderate choices of $m$ naturally satisfy this condition. When $L$ is unknown, a practical approach is to use moderate values of $m$ and, if desired, examine results across several choices, as is commonly done in classical portmanteau testing. We have clarified these points in the revision.
>
>
> > 4 The paper ultimately favors SIP 2, yet it does not sufficiently explain why the two statistics, $T^{SIP}_{n,1}$ and $T^{SIP}_{n,2}$ , exhibit different finite-sample performance, nor does it clarify their respective domains of applicability or the sources of their differences in stability. A more in-depth analysis is warranted. In particular, the authors should more explicitly articulate the differences in the construction of the two statistics and their corresponding statistical interpretations, rather than relying primarily on empirical comparisons.
>
> **Response:**
> We thank the reviewer for this helpful suggestion.
>
> The primary distinction lies in how the variance component $\gamma_0$ and the associated parameter $w$ are estimated. SIP1 is based on a direct quadratic-form estimator $\hat{\gamma}_0$, which remains consistent under mild dependence conditions (e.g., $m$-dependence), making it more robust to model misspecification. In contrast, SIP2 employs estimators $(\hat{\alpha}, \hat{\beta})$ derived from a regression formulation, which are tailored to the null hypothesis and typically exhibit lower variance. This leads to a trade-off between robustness (SIP1) and stability (SIP2), particularly when $m$ is moderately large.
>
> We have expanded the discussion comparing SIP1 and SIP2, both in terms of their construction and empirical behavior. In particular, we clarify that the primary difference lies in the estimation of the variance component, and that SIP2 provides more stable performance when $m$ is large. This is now more explicitly discussed in the revised manuscript.
>
> In terms of power, neither method uniformly dominates the other, and their relative performance depends on the underlying dependence structure. Our expanded numerical studies indicate that both procedures perform similarly in most settings, with SIP2 often showing slightly improved finite-sample stability for larger values of $m$. We have revised the manuscript to make these distinctions more explicit.

---

> ### Author Response · Authors · 2026-04-13
> **Response to Reviewer G6sv (part 3)**
>
> > 5 Section 3.3 would benefit from a clearer definition of the testing target. The overarching objective of the paper is to test whether the process is free of serial correlation, i.e., $\rho_h=0$ for all $h>0$
> . However, for a given m, the formal hypothesis actually tested in the paper is $P_{m+2}\gamma_{m+2}$
> . It is recommended that the authors explicitly distinguish, in Section 3.3, between the overall research objective and the formal statistical hypothesis under a fixed m, and clarify the relationship between the two. This would help avoid potential confusion where readers may directly equate “testing all autocorrelations being zero” with “testing $P_{m+2}\gamma_{m+2}$.”
>
> **Response:**
> We have added a paragraph in Section 3.3 to explicitly distinguish between the overall goal of testing $\rho_h = 0$ for all $h > 0$ and the formal hypothesis tested for a fixed $m$, namely $P_{m+2}\boldsymbol{\gamma}_{m+2} = 0$.
>
> > 6 The manuscript still contains several noticeable typographical errors and grammatical issues, and a thorough proofreading is recommended to ensure consistency and clarity throughout. For example:
> (1) in “In this subsection. we investigate ...”, the punctuation is incorrect;
>
> (2) in “if $m$ an integer such that ...”, a linking verb is missing;
>
> (3) there are occasional spelling errors in table captions and the main text.
>
> **Response:**
> Thank you so much for catching the errors, We have carefully proofread the manuscript and corrected typographical and grammatical errors throughout.

---

### Decision · Action_Editor_2dLH · 2026-04-23

**Recommendation:** Accept as is

**Comment:**

This manuscript presents a original and well-executed solution to a fundamental problem in time series analysis. The authors identify a critical failure point of classical portmanteau tests under frequent mean shifts and develop a sophisticated yet practical inferential framework to address it. The work is theoretically sound, empirically robust, and offers clear value to a wide audience.

Two expert reviewers assessed the manuscript, and both recommended acceptance.  I recommend Accept without further revisions.

**Audience:**

Yes, the findings are highly relevant and of significant interest to the broad readership of SLADS_Section_B.

The paper makes both a deep methodological contribution to time series analysis and provides a powerful new tool for an important contemporary application. The work will appeal to multiple segments of the journal’s audience, inclluding methodologists, practitioners
as well as applied Scientists.

**Claims And Evidence:**

Yes, the claims are fully supported by accurate, convincing and clear evidence.

The authors make three core claims: (1) classical portmanteau tests fail under frequent mean shifts; (2) the proposed SIP test effectively controls type I error in this setting; and (3) the method is highly useful for real-world applications, particularly nanopore sequencing data.

These claims are substantiated through:
(1) Theoretical Rigor: The paper provides rigorous proofs for constructing shift-immune quadratic forms and establishes the asymptotic null distribution of the test statistic (Theorems 1–3).
(2) Comprehensive Simulations: Extensive numerical studies demonstrate that the SIP test controls type I error where traditional Box-type tests and alternative bootstrap methods fail completely (empirical rates of 1.00). Power analyses across diverse dependence structures (MA, AR, ARMA, bilinear) and mean configurations further validate the method.
(3) Real-Data Validation: Analysis of 900 nanopore sequencing reads successfully detects subtle yet significant autocorrelations that classical ACF plots miss, directly addressing a conjecture in the field and offering a compelling practical application.